

# Going beyond the Flood Insurance Rate Map: insights from flood hazard map co-production

Adam Luke[1], Brett F Sanders[1,2], Kristen Goodrich[3], David L Feldman[2], Danielle Boudreau[4],
Ana Eguiarte[4], Kimberly Serrano[5], Abigail Reyes[5], Jochen E Schubert[1], Amir AghaKouchak[1],
Victoria Basolo[2], and Richard A Matthew[2]

[1]Department of Civil and Environmental Engineering, University of California, Irvine, CA, USA
[2]Department of Urban Planning and Public Policy, University of California, Irvine, CA, USA
[3]School of Social Ecology,University of California, Irvine, CA, USA
[4]Tijuana River National Estuarine Research Reserve, Imperial Beach, CA, USA
[5]Sustainability Initiative, University of California, Irvine, CA, USA

*Correspondence to:* Adam Luke (aluke1@uci.edu)

**Abstract.** Flood hazard mapping in the United States (US) is deeply tied to the National Flood Insurance Program (NFIP). Consequently, publicly available flood maps provide essential information for insurance purposes, but do not necessarily provide relevant information for non-insurance aspects of flood risk management (FRM) such as public education and emergency planning. Recent calls for flood hazard maps that support a wider variety of FRM tasks highlight the need to deepen our understanding about the factors that make flood maps useful and understandable for local end-users. In this study, social scientists and engineers explore opportunities for improving the utility and relevance of flood hazard maps through the co-production of maps responsive to end-users' FRM needs. Specifically, two-dimensional flood modeling produced a set of baseline hazard maps for stakeholders of the Tijuana River Valley, US, and Los Laureles Canyon in Tijuana, Mexico. Focus groups with natural resource managers, city planners, emergency managers, academia, non-profit, and community leaders refined the baseline hazard maps by triggering additional modeling scenarios and map revisions. Several important end-user preferences emerged, such as 1) legends that frame flood intensity both qualitatively and quantitatively, and 2) flood scenario descriptions that report flood magnitude in terms of rainfall, streamflow, and its relation to an historic event. Regarding desired hazard map content, end-users' requests revealed general consistency with mapping needs reported in European studies and guidelines published in Australia. However, requested map content that is not commonly produced included: 1) standing water depths following the flood, 2) the erosive potential of flowing water, and 3) *pluvial* flood hazards, or flooding caused directly by rainfall. We conclude that the relevance and utility of commonly produced flood hazard maps can be most improved by illustrating pluvial flood hazards and by using concrete reference points to describe flooding scenarios rather than exceedance probabilities or frequencies.

## 1  Introduction

Management of flooding is a major societal challenge that is only expected to worsen in the future due to several trends including population growth and urbanization (Sundermann et al., 2014), sea level rise (Hallegatte et al., 2013), intensification




of precipitation extremes (Lenderink and Van Meijgaard, 2008; Coumou and Rahmstorf, 2012), and the compounding effects of sea level rise and terrestrial flooding (Moftakhari et al., 2017). Insured losses from natural disasters have been increasing globally (Munich Re, 2005), largely from the growing exposure and value of vulnerable assets (Bouwer, 2011). Losses from hurricanes and floods in the United States (US) have tripled over the past fifty years (Gall et al., 2011), and the National Flood

Insurance Program (NFIP) is operating at a deficit of about $1 billion annually with a debt of over $20 billion owed to the US treasury before considering insured losses from the 2017 hurricane season (Pasterick, 1998; Brown, 2016). In fact, properties insured by the NFIP represent the second largest liability of the US federal government after the Social Security program (Gall et al., 2011).

The American Society of Civil Engineers (ASCE) has called for a national strategy to address the escalation of flood losses

and threats to public safety, but reports that the US public and policy makers have been unwilling to take action despite major hurricanes such as Katrina and Sandy (Traver, 2014). The ASCE directive aligns with a global paradigm shift in management philosophy away from *flood control* and towards *flood risk management*. Flood risk management (FRM) refers to a portfolio of approaches for reducing risk that is not limited to controlling flood waters with engineered structures, but also includes effective land use planning, emergency response, and personal preparedness. Importantly, FRM accepts that absolute protection is not

possible. Comprehensive FRM reduces the reliance on engineered flood defenses, which is of paramount importance in the US due to the marginal condition of levees and lack of federal resources available for maintenance and necessary upgrades (Traver, 2014). Studies have shown that robust FRM does indeed lead to significant reductions in fatalities and monetary losses (Kreibich et al., 2017, 2005), however Traver (2014) and Merz et al. (2007) both report that effectively implementing FRM relies on stakeholders who understand their exposure and also have access to tools that are useful for managing personal,

household, and community risks.

Flood hazard maps are the most commonly used tool for flood risk communication and management. In the European Union (EU), member countries are under a mandate to develop national flood hazard maps, and general guidelines for meeting end-user needs have been developed based on participatory processes (Meyer et al., 2012; Hagemeier-Klose and Wagner, 2009; Martini and Loat, 2007). Guidelines reflect the varying needs of different end-users for different types of information, as well

as the need for context-sensitive information. For example, Meyer et al. (2012) present distinctions between the mapping needs for strategic planning personnel, emergency management personnel and the public, and show that geographical factors (e.g., mountains, polders) influence the need for velocity data.

In the US, flood mapping is tied to the NFIP and the resulting Flood Insurance Rate Maps (FIRMs) delineate the spatial extent of inundation with a 1% and 0.2% annual exceedance probability (AEP). As a vehicle designed to administer an insurance

program, the FIRM provides essential information for insurance purposes. Properties with federally backed mortgages located within the 1% AEP floodplain are required to purchase flood insurance, while the flood elevations associated with the FIRM are used for insurance underwriting. However, the binary "in or out" floodplain designation by the FIRMs' *thin grey lines* have been criticized for presenting flood risk as definitive and therefore discouraging important flood hazard discourse (Soden et al., 2017). Burby (2001) also suggests that the effectiveness of the NFIP is limited because FIRMs lack information necessary to

integrate flood hazard considerations into local planning. The Federal Emergency Management Agency (FEMA) has recently



expanded its mapping efforts through the Risk MAP program (FEMA, 2014), which produces "non-regulatory" flood hazard data such as depth, velocity, and exceedance probability grids in addition to the standard FIRM (FEMA, 2016). However, the availability of non-regulatory flood hazard data is limited, and the system is not configured to align mapping products with context-sensitive needs for decision-making. In fact, there is a need for US-centric studies and guidelines for producing maps

that are useful for a variety of FRM tasks, i.e., making flood hazard data useable for local end-users across vast hydrologic and social conditions. Flood mapping technology has evolved rapidly over the past decade with modern two-dimensional (2D) hydraulic flood modeling software, computing systems, and increasingly available high quality data to produce point-wise flood hazard information including flood depths, velocities, flood forces and shear stresses (Sanders, 2017). Hence, a key issue is making this advanced and complex modeling output useable in FRM.

Producing useable scientific information for non-expert end-users is quite challenging, however, as demonstrated by the climate science community (Dilling and Lemos, 2011). Useable scientific information (or flood hazard maps in this study) must bridge the gap between what the producers of scientific knowledge deem useful and what is actually helpful in practice. If scientists produce information absent of end-user input, then the produced knowledge is not always applicable to solving problems. Conversely, if end-users set the agenda, end-users may require information that is not possible to produce or weak

scientifically. The co-production of scientific knowledge attempts to avoid these issues through an iterative process involving domain experts and end-users. Studies of co-production suggest information must be "end-to-end" useful - applicable to the needs of many users, adaptable for disaster planning as well as mitigation, and able to incorporate local knowledge of threats and hazards (Agrawala et al., 2001; Feldman and Ingram, 2009). Dilling and Lemos (2011) reported that nearly every example of the successful use of climate knowledge resulted from iteration between the producers and users of scientific information,

while the process of iteration itself can also uncover new uses that have not been previously identified. Herein, we recognize the lessons learned from the climate science community by engaging in the co-production of flood hazard maps.

Within the study of engaged scholarship there is a broad spectrum of knowledge "co-production", depending on the degree of iterativity and duration of scholarly engagement. In this paper, we report a single iteration of a broader co-production effort known as FloodRISE, which aims to promote resilience to flooding in Southern California through continued, meaningful

efforts of engagement. The goal of the present study is to both deepen understanding about the factors that make flood hazard maps useable at the local scale and to expand the applications of flood hazard mapping via lessons learned from the co-production of flood hazard maps. In order to meet this objective, an interdisciplinary team of engineers and social scientists developed a set of baseline flood hazard maps for the stakeholders of the Tijuana River Valley in California, US and Los Laureles in Baja California, Mexico (MX). Following baseline hazard mapping by engineers at the two sites, focus groups

were held with a diverse group of end-users comprised of 52 local professionals and community members. The focus groups were designed to understand 1) how to improve the clarity and utility of the baseline hazard maps and 2) how to re-configure the hydraulic models to produce relevant data and useful maps for the communities. In addition to many map revisions, several original flood hazard maps were produced as a direct result of this iteration of knowledge co-production.

The paper continues with a description of the two study sites involved in the co-production of flood hazard maps (Section 2).

In Section 3, we present the baseline (pre-focus group) flood hazard maps produced for each site. Detailed descriptions of the



methods used to produce each hazard map are included in the Appendices. Section 4 outlines the implementation and design of our end-user focus groups. We present the results of the end-user focus groups in Section 5.1-5.2, and the new hazard maps that resulted from the focus groups in Section 5.3. This paper concludes with a discussion of these Sections in the context of previous studies and current flood mapping practice in Europe, Australia, and the US.

## 5    2    Study Site Descriptions

The Tijuana River Valley (TRV) and Los Laureles (LL) communities were selected for the co-production exercise due to their on-going FRM challenges. The TRV is adjacent to the US/MX border and forms the Southwestern most corner of the contiguous US (Fig. 1A-B). The Tijuana River watershed encompasses 4530 $km^2$ (1750 $mi^2$) and is characterized by four major water-supply reservoirs which control about 75% of storm water flow. Due to its location at the mouth of the Tijuana

River, the TRV is prone to flooding from Tijuana River discharges, storm tides, ocean swell, and intermittent flows from LL and Smugglers Gulch (Fig. 1A, Appendix A1). The concrete-lined Tijuana River Channel was designed to convey spillway discharges from upstream reservoirs safely through downtown Tijuana. Sediment accumulation has reduced the capacity of the Tijuana River channel near the US/MX border and requires dredging to maintain its design conveyance. The TRV has rural housing, an equestrian (business) presence, and government facilities, but land use in the TRV is primarily preserved for natural

habitats or agricultural uses (TRNERR, 2010). Land use in the TRV contrasts the Mexican side of the border where population density is high due to the presence of colonias and formal settlements.

     LL is a canyon community of Tijuana located south of the US/MX border in the LL catchment (Fig. 1C). The catchment is relatively small compared to the Tijuana River Watershed, and flooding is caused by locally intense precipitation. Flows from LL enter the TRV from the south after passing through a small culvert at the watershed outlet. The culvert is fed by a network

of concrete lined storm-water channels, which is continually expanding to prevent stream bed erosion and protect adjacent housing developments. The network of concrete lined channels and expanding urban development has increased the discharges that must be conveyed by the culvert. In both study sites, flooding is a known and recurring issue. Ponding has been observed behind the LL culvert during rainstorms, and a culvert blockage lead to severe flooding and subsequent evacuations in LL and southern Imperial Beach (personal communication, March, 2016). In the TRV, flooding occurs when the upstream reservoirs

reach capacity and are forced to open their spillways. Spillway discharges have occurred seven times from 1940 to the present (IBWC, 2006). Indeed, hazard maps which only delineate at risk areas have limited use for locals; stakeholders already know the TRV and LL are flood prone. Hazard maps that depict more than the traditional 100 year flood extent are, however, scarce.




**Figure 1.** A) Tijuana River Valley and relevant features. The Valley is bounded by the City of Imperial Beach to the North, the City of San Diego to the East, and the City of Tijuana to the South. B) Tijuana River Watershed and broader geographical context. Notice the international aspect of the Watershed; about one third of the Watershed area is within the US, and the rest is within Mexico. C) Los Laureles community and Los Laureles main channel. The culvert is also shown, which conveys storm water discharges from the channelized section of the Los Laureles stream network.

## 3 Baseline Flood Hazard Maps

Baseline flood hazard maps were produced to demonstrate to end-users the range of flood hazard data that can be produced with modern methods, and to stimulate discussion about improvements upon the hazard maps and desired content. At this point, it is helpful to define "flood hazard" as the physical and probabilistic characteristics of floods. The baseline flood hazard maps therefore depict the intensity of spatially varying properties of floods, such as the depth of flooding for a specified probability. We explicitly define flood hazard to distinguish between maps of *hazard* and *vulnerability*. Merz et al. (2007) defines vulnerability as the combination of loss susceptibility and damage potential. Thus, flood vulnerability maps show what





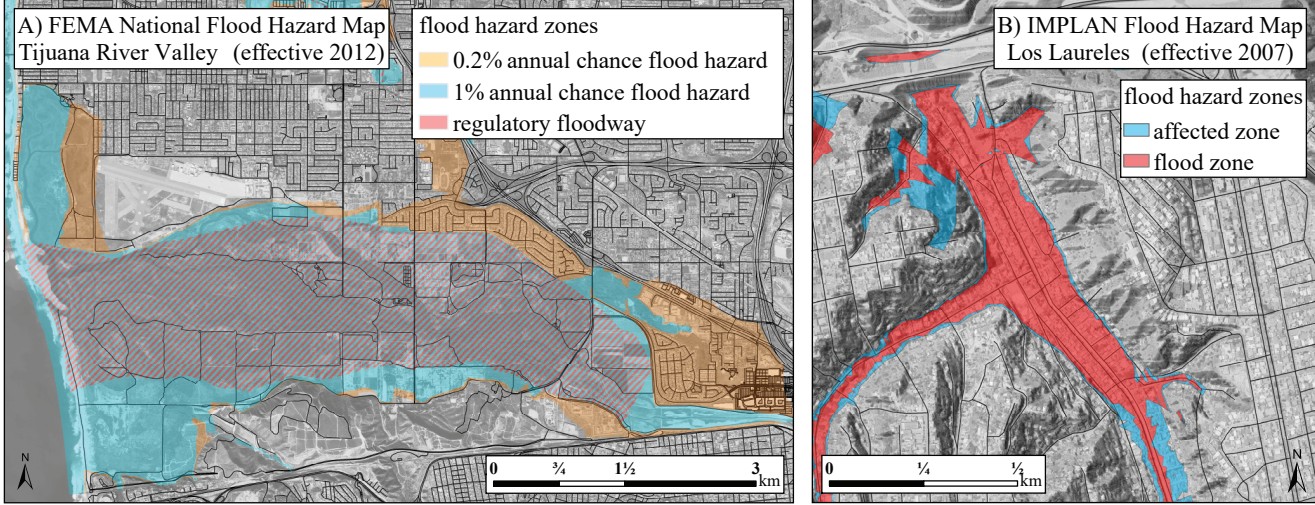

**Figure 2.** A) Federal Emergency Management Agency Flood Insurance Rate Map for the Tijuana River Valley. The *orange* area represents the flooding extent associated with the 0.2% annual exceedance probability (AEP) flood, while the *blue* area represents the flooding extent of the 1% AEP flood. The *red* hashed area delineates the regulatory floodway, where development must not increase the designated base flood elevation by more than 0.3 m (1 ft). B) IMPLAN flood hazard map for the Los Laureles subbasin. The *red* and *blue* zones represent inundation extent for different exceedance probability flooding events. However, details regarding mapping methodology and precise hazard zone explanations were not available.

could be affected by floods and by how much what is affected could be damaged. We restrict our analysis to flood hazard maps because producing vulnerability maps generally requires data that are unavailable in the absence of extensive surveys. In this study, end-users analyzed maps depicting six different flood hazards: depth, force, exceedance probabilities, dominant causes, durations, and extents. The proceeding section presents the hazard maps analyzed by the end-users most applicable to other
sites, i.e. we do not present all of the baseline hazard maps herein. The hazard maps we produced were analyzed together with the publicly available hazard maps.

Figure 2 shows the publicly available hazard maps, which will serve as the baseline flood *extent* hazard maps in this study. The FEMA FIRM is shown in Fig. 2A, while Tijuana's Instituto Municipal de Planeacion (IMPLAN) Flood Hazard map for LL is shown in Fig. 2B. Both hazard maps depict the inundation extent associated with different AEPs, while the FEMA FIRM
also depicts a regulatory floodway. Development in the regulatory floodway is prohibited unless the proposed structure will not increase the base flood elevation by more than 0.3 m (1 ft). Notice also that the flood hazard is described by its associated annual probability, which is consistent with official FEMA mapping legends and terminology. The IMPLAN Flood Hazard map is similar; areas at risk of flooding are shown, but flooding extent is the only characteristic depicted. Flooding extent is also the most commonly mapped hazard in Europe, but many European countries also provide maps of flood depth (Moel et al.,
2009; Nones, 2017).



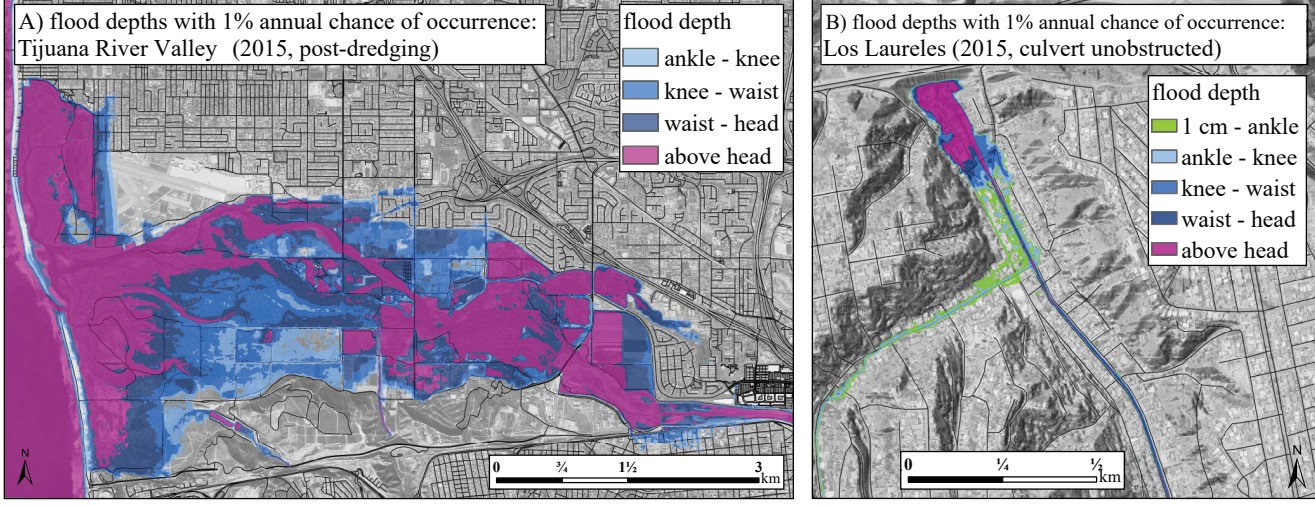

**Figure 3.** A) 1% annual exceedance probability (AEP) flood depths in the Tijuana River Valley. Mapped flood depths result from either storm tides, canyon flows, or Tijuana River discharges (Appendix A1). The elevation data reflects 2015 topography, after dredging of the Tijuana River channel near the US/MX border. B) 1% AEP flood depths in Los Laureles. Mapped flood depths are caused by streamflows from local precipitation, assuming that the culvert was unobstructed during the duration of the flood. The depth ranges are $0.11 - 0.45$ m for ankle to knee, $0.45 - 1.0$ m for knee to waist, $1.0 - 1.69$ m for waist to head, and greater than $1.69$ m for above head depths.

Figure 3A shows the analyzed flood *depth* map for the TRV, while the depth map for Los Laureles is shown in Fig. 3B. Both maps characterize depths using a body scale, and depict depths associated with 1% AEP events. The body scale is based on the average person height reported by Fryar et al. (2012), with the body part thresholds defined using the 7.5 heads rule from the field of artistic anatomy (Richer, 1986). We use a body scale to contour flood depths with the intention of producing flood hazard data that is more relatable to end-users. The clear advantage of providing maps depicting the depth of flooding is that potentially unsafe areas during extreme events within the floodplain are easily identified. Fig. 3 also demonstrates our (pre-focus group) approach for communicating the mapped hazard. Specifically, we elected to describe flood hazards by their annual probabilities and contour intensity of the hazard with a qualitative scale. Data shown on the map other than the flood hazard was limited to allow end-users to focus requests and revisions on the hazard data itself. All of the baseline hazard maps were presented and described in a similar manner.

Figure 4 shows the analyzed flood hazard maps which incorporate flow velocity information, where the flow velocity, $v$, multiplied by flood depth, $h$, is contoured using a scale reflecting the strength or "force" of the flood waters. We use $vh$ as a proxy for the force of the flood waters because thresholds for toppling people and moving cars have been reported in terms of $vh$, or discharge per unit width (Xia et al., 2014, 2011). A $vh$ criterion was also found suitable for predicting structural damage to homes (Kreibich et al., 2009; Gallegos et al., 2012), so $vh$ can be used to describe a range of hazardous conditions. Although the "force" map is more precisely described as a flood discharge per unit width map, we use the word force for




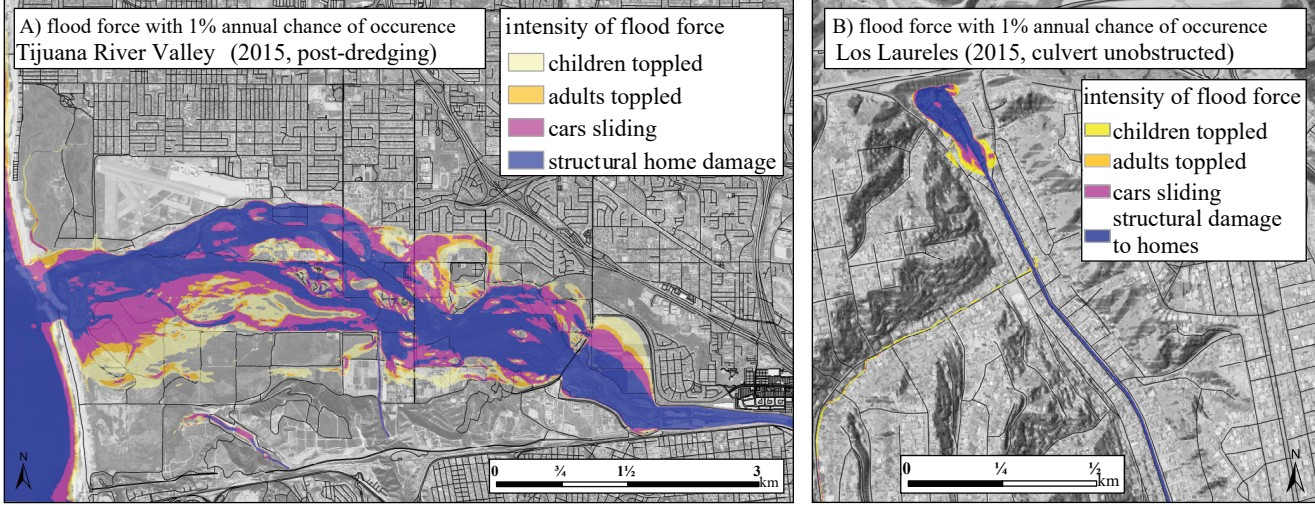

**Figure 4.** A) Flood force with 1% annual exceedance probability (AEP) in the Tijuana River Valley. B) Flood force with 1% AEP in Los Laureles. We use $vh$ as a *proxy* for the flood force, since contours are specifically different thresholds of $vh$: children toppled at 0.4 m$^2$/s, adults toppled at 0.65 m$^2$/s (Xia et al., 2014), cars sliding at 0.8 m$^2$/s (Xia et al., 2011), and structural damage beginning at 1.5 m$^2$/s (Kreibich et al., 2009).

simpler communication. By relating the intensity of $vh$ to the vulnerability of people, cars, and homes, the map could be used for strategic land use planning or emergency response. We note that Australian flood studies produce information describing the severity of $v$ and $h$, and recommend that hazard zones are defined by $vh$ thresholds (AEMI, 2013).

The baseline hazard maps were associated with the 1% AEP event for consistency with the usual presentation of flood hazard information, although each hazard map presented thus far could be produced for an event more or less likely than 1% AEP. Hazard maps can also display multiple AEP events on a single map. This is accomplished by contouring the AEP of a particular hazard threshold, rather than contouring the intensity of a flood hazard with a specified AEP. Figure 5 presents the analyzed maps which contour AEP rather than intensity of depth or force. The AEP of ankle depth flooding in the TRV is shown in Fig. 5A, while Fig. 5B displays the AEP of flood forces strong enough to topple children in LL. AEP maps of either $h$ or $vh$ thresholds are attainable by simulating multiple floods and combining the results on a single map. The maps shown in Fig. 5 can be produced using a wide variety of hazard thresholds not limited to ankle depth flooding or children toppled. Here, mapping exceedance probabilities of a specific hazard threshold quantifies and communicates the likelihood of a particularly negative consequence of flooding. However, the AEP map is difficult to produce (Appendix A3) and presents highly technical information. Thus, it may present challenges in communicating with end-users.

The maps presented in Fig. 2-5 can be produced for a wide variety of locations and are not limited to site specific hydrologic conditions. We do not present the site specific baseline hazard maps produced for the end-users in this paper; however, the site specific hazard maps illustrating dominant drivers and duration of inundation can be viewed online following the links in





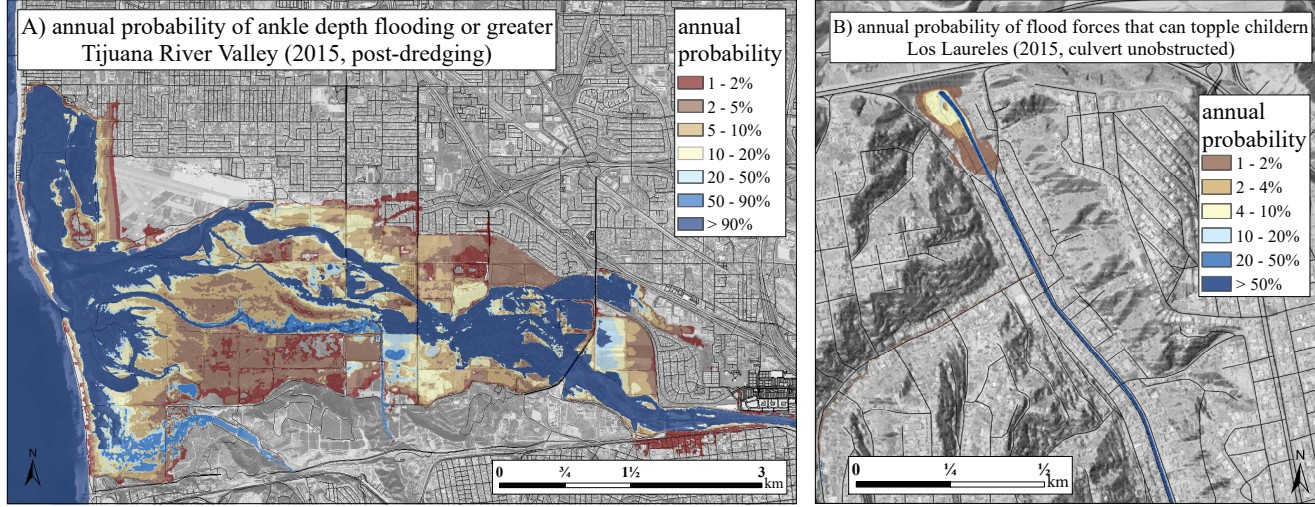

**Figure 5.** A) Annual exceedance probability (AEP) of ankle depth flooding in the Tijuana River Valley. Contours represent the annual probability that either storm tides, canyon flows, or Tijuana River discharges can cause flooding that exceeds ankle depth. B) AEP of flood forces that can topple children in Los Laureles. Here, contours represent the annual probability that streamflow will be intense enough to topple children. The ankle depth threshold was defined as 0.11 m, and the children toppled threshold was defined as 0.4 m$^2$/s.

the *data availability* section. The hazard maps presented in Fig. 2-5 exemplify the range of flood hazard data presented to the end-users and our original approach for communicating the mapped hazard. We note that Fig. 2-5 are not identical to the maps analyzed by the end-users since they were adapted for presentation format herein. Specifically, legend placement and fontsize is slightly different in Fig. 2-5 from the maps analyzed by the end-users, while Fig. 2-5 also do not include street and landmark labels that were included in the baseline maps. However, the legend content and text, map titles, hazard data, base maps, and color schemes are identical between maps analyzed by end-users and Fig. 2-5. The cartographic design of the baseline hazard maps generally followed recommendations for designing flood risk maps provided by Fuchs et al. (2009).

## 4 Stakeholder Focus Groups

Four focus groups were held at each study site, which were distinguished by job-function related to FRM. The TRV focus groups included 22 total participants, with six representing public works professionals and city planners, eight representing emergency managers, five representing natural resource managers, and three categorized as non-governmental organizations or community. The LL focus groups included 33 participants total, with nine individuals representing an academic affiliation, six representing government, eleven representing emergency managers, and seven representing non-governmental organizations or other community members. Participants of each focus group were recruited through personal communication and referral sampling. It was not possible to strengthen reliability of the focus groups by replicating cohorts because a limited number of individuals are involved with FRM at the study sites. Thus, by limiting our focus groups to individuals who play a role in



local FRM, we strengthened the study cohort by including the most relevant participants based on job function. Representation was not identical at the two sites because 1) governmental and organizational structures vary between the two countries and 2) participants were chosen based upon their importance to decisions related to FRM in the respective sites.

The TRV and LL focus groups were each two and a half hours in length, during which a facilitator elicited information
regarding the focus group's perception of the baseline flood hazard maps. A large format hardcopy of each map was distributed to all participants along with a glossary that described baseline map terms. After individually examining the map, the facilitator would ask the engineer who produced the map to explain the hazard map and establish a common understanding. Participants were able to ask the engineer questions, while the facilitator would ask questions to test understanding. Data was generated throughout the focus groups by the facilitator adhering to a script with questions specific to each baseline hazard map. For
example, the facilitator would ask participants about their opinions of the indicators used in the map legends, the utility of the mapped hazard data, and information they would like to see depicted but was not included. The complete facilitation plan is available in the supplemental materials. This process repeated for each of the baseline hazard maps.

Following the presentation and discussion of the baseline hazard maps, survey data was collected. The surveys were designed to elicit information that would be useful for re-configuring the hydraulic models and producing flood hazard data relevant to
the focus group participants. Results from the final "exit" survey addressed several important aspects of producing relevant flood hazard data: end-user job function, planning time frames of interest, flood return periods of interest, flood drivers of interest, relevant FRM strategies, and environmental conditions of interest. Each question in the survey was followed by a range of options, and the focus group participants were prompted to select a single option in response to each question. The full exit survey is also provided in the supplementary information.

Data was collected throughout the focus groups by recording conversations and reinforced by note-taking. Furthermore, transcripts were prepared of all conversations based on audio recordings. Transcripts were translated into English then analyzed using open coding to identify general themes and concepts followed by axial coding (categorization) and identification of patterns and relationships among the concepts (Saldaña, 2015; Feldman, 1995). Transcripts of the focus groups were analyzed independently using codes including "Requested Map Revisions" and "Requested Map Scenarios" specifically to determine
end-user expressed (1) requested improvements to baseline maps and (2) desirable flood hazard maps. "Requested Map Revisions" included requests and inquiries related to the map legend, units, and contextual information. Information that was useful for re-configuring the hydraulic models (new scenarios) and producing new hazard maps were categorized as "Requested Map Scenarios." Flood mapping scenarios that were specifically mentioned or requested by participants were verified by independent analysis. Data obtained from the exit surveys were also categorized as "Requested Map Scenarios."

## 5  Results

### 5.1  Requested Map Revisions

Table 1 summarizes the requested map revisions of the end-users from both sites. The requested revisions generally fell into three categories: requests for the hydrologic/meteorologic conditions of the mapped hazard, further clarification of the map



**Table 1.** Summary of the end-users' requested map revisions. These specific requests were identified through transcript content analysis.

| | Tijuana River Valley Stakeholders | Los Laureles Stakeholders |
|---|---|---|
| **requested hydrologic/meteorologic information:** | - amount and duration of precipitation<br>- volumetric flow rate | - amount and duration of precipitation<br>- flooding mechanism (cause)<br>- conditions related to previous flood |
| **legend requests/comments:** | - quantify legend thresholds<br>- scientific units<br>- erosion scale<br>- children toppled threshold not appropriate | - quantitative information<br>- scientific units<br>- relate flooding to erosion<br>- relate velocity to infrastructure damage<br>- non-technical language |
| **additional geospatial data requests:** | - access roads<br>- river channel<br>- levees and dikes<br>- landmarks<br>- park trails<br>- sediments and vegetation | - access roads<br>- channels<br>- population/demographics<br>- sediment basins<br>- topography<br>- aerial photo with year<br>- locations of dwellings and shelters |

legends, or requests for additional geospatial data to be shown on the map. The requests presented in Table 1 were specifically mentioned by participants and confirmed by discussions recorded in the transcripts.

There were several important requested map revisions that were common to the end-users of both sites. First, end-users were particularly interested in the amount of rainfall or streamflow that caused the flood hazard on the map. The amount and duration of precipitation leading to the mapped hazard was often requested for both sites, while TRV end-users requested the flow rate of the Tijuana River associated with the hazard maps. This illustrates the desire to relate the mapped hazard to information that is available in real time or other publicly available information. LL end-users also asked how the mapped scenario related to previous flooding events and noted that the precise frequency of the storm is not necessary for many users. Notice that the baseline hazard maps were described by the probability of the mapped hazard only (Fig. 3 - 5), and not by the conditions leading to the mapped flood hazard such as the amount of rainfall.

Second, participants of both sites were interested in the quantitative or scientific units of the hazard legends. In general, the participants found the qualitative legends helpful and informative, but clarification was needed to explain the basis for the qualitative thresholds. For example, the "children toppled" hazard criteria (Figure 4) was received with mixed reactions. Several participants were confused by the distinction between hazardous conditions for children versus adults, which required additional explanation, and others viewed this criteria as alarmist. A participant representing public works professionals said: "I think [children toppled] is an alarming metric. I don't know if it would be useful for my city work... maybe a different way of identifying intensity would be more useful." Another participant representing natural resource managers noted: "I would be





interested in knowing the velocity or what the force impacts are. For what I do, I can't write in an EIS [Environmental Impact Statement] that a child would be toppled in this area, so that wouldn't work for me." End-users representing natural resource managers were generally more interested in relating the velocity information to the erosive potential of the flowing water, and several participants recommended an erosion scale based on velocity and land cover data. On the other hand, end-users

representing emergency managers did find the metrics in the force map (Fig. 4) useful.

Emergency managers noted that the flood intensity descriptors of the force map would help them decide where to allocate resources during a flood based on the depicted severity, while the "cars sliding" metric would help determine which roads would be inaccessible during an extreme event. Emergency managers also said the force map would help them prepare for hazardous debris flow during a flood: "Let me tell you why this [force map] is really important... what I'm really concerned

about down in the Valley is...anything that's a hazard in the river that's actually moving. If I see a barn, and I know that its structural home damage in this area, I'll be a little more on alert because... hey those barns and stuff, they're gone. And so now I know there's more things in the river that not only can create a hazard to me but that can also adjust the flow of the river as well." Indeed, participating emergency responders were primarily interested in the relative severity of the flood hazard within the floodplain.

Lastly, end-users requested a wide variety of additional geospatial data to provide additional context, i.e, information beyond the illustrated flood hazard. The common geospatial data requests between the two sites included depictions of access roads, the river channels, and important flood control infrastructure. Of course, the relevant geospatial data will depend on the end-user and site specific characteristics, but many end-users requested the locations of access roads and river channels.

### 5.2   Requested Map Scenarios

Transcripts were examined to extract information that was useful for re-configuring the hydraulic models for new scenarios. This information includes environmental conditions of interest, relevant magnitudes and types of flooding events, and requested flood hazard data (model output). Re-configuring and re-running the models involves changes to model inputs such as a topographic data and boundary conditions. Table 2 shows the data we collected from the transcripts that were relevant for producing the new flood hazard maps. In addition to the focus group transcripts, data from the exit surveys were used to

configure relevant flood modeling scenarios. Figure 6 shows the results of the exit survey as a percentage of the participants' responses. We note that these results are not statistically significant nor representative of all end-users of flood hazard data, but they were useful for complementing the qualitative information obtained from the transcripts. Comparison between the exit survey results (Fig. 6) and the modeling requests documented in the transcripts (Table 2) reveal several important stakeholder preferences and are considered highly relevant to the specific end-users in this study.

Regarding the magnitude of the mapped flooding event, end-users were interested in events more frequent, or smaller in magnitude, than the 100 year (or 1% AEP) flood. TRV and LL end-users specifically asked to see hazards associated with more frequent events (Table 2), and when asked "What return period is most useful to be mapped?", more than half of end-users from both the TRV and LL focus groups selected return periods of 20 years or less. The 100 year flood was still considered relevant, however end-users generally considered the hazards of frequent floods more useful for day-to-day decision making. One



**Table 2.** Summary of information used to re-run the hydraulic models and produce new hazard maps (flood mapping requests). These specific requests were identified through transcript content analysis.

| | Tijuana River Valley Stakeholders | Los Laureles Stakeholders |
|---|---|---|
| **relevant flood magnitudes and drivers** | - historic floods | - historic floods |
| | - more frequent than 100 year/ 1% AEP | - more frequent than 100 year/ 1% AEP |
| | - storm water runoff | - pluvial flooding |
| | - nuisance flooding/ non-extreme events | |
| | - wave overtopping (Imperial Beach) | |
| **environmental conditions[a] of interest** | - blockages/obstructions | - blockages/obstructions |
| | - worst case flooding/infrastructure fails | - worst case scenario |
| | - different channel capacities | - future channelization |
| | - different channel locations | - future land use |
| | - early season/late season | |
| | - sea level rise | |
| **requested flood hazard data/model output** | - output related to erosion potential | - output that predicts erosion potential |
| | - output that depicts standing water | - output based on forecasted rain |
| | - duration of flooding event | - duration of flooding |
| | - velocity of flood waters | - maximum velocity of flood |
| | - real-time data | |
| | - suite of maps related to different rainfall depths | |

[a]"Environmental conditions" are defined as the physical conditions of study area or the state of climate related variables during the simulated flood.

participant noted: "We often focus a lot on the 1% [flood] because of FEMA and things like that, but in terms of what people are experiencing right now, they're noodling around with like 5 or 10 year events which is what's causing them problems... hitting those frequencies that people are more likely to see, that would be a valuable communication tool."

Another important finding that was particularly evident from the LL focus groups was interest in the hazards of pluvial flooding, or flooding caused by storm water runoff. Indeed, twice as many LL end-users selected "pooled rainfall" compared to "excessive streamflow" as the most useful flood driver to be mapped. This interest was documented in the transcripts for both sites, i.e., TRV end-users noted that storm water runoff can cause flooding in neighborhoods just north of the River Valley. One of the TRV end-users described how nuisance flooding associated with storm water is noticeably absent from publicly available maps: "We have areas in the city that experience nuisance flooding, and even though it doesn't show up on the FIRM maps as an area of special flood hazard, ...you almost have to find out from property owners in there, knock on their doors, [and ask] do you ever get floods here? [owners respond] 'Oh yeah all the time'." LL end-users were also concerned with flooding caused by storm water runoff, but they were more interested in flood hazards caused by extreme rainfall events along steep, canyon terraces that are not adjacent to the storm water channels. Mapping the hazards associated with direct rainfall is considerably different from traditional floodplain mapping, whereby the floodplain is delineated by routing discharges that exceed channel



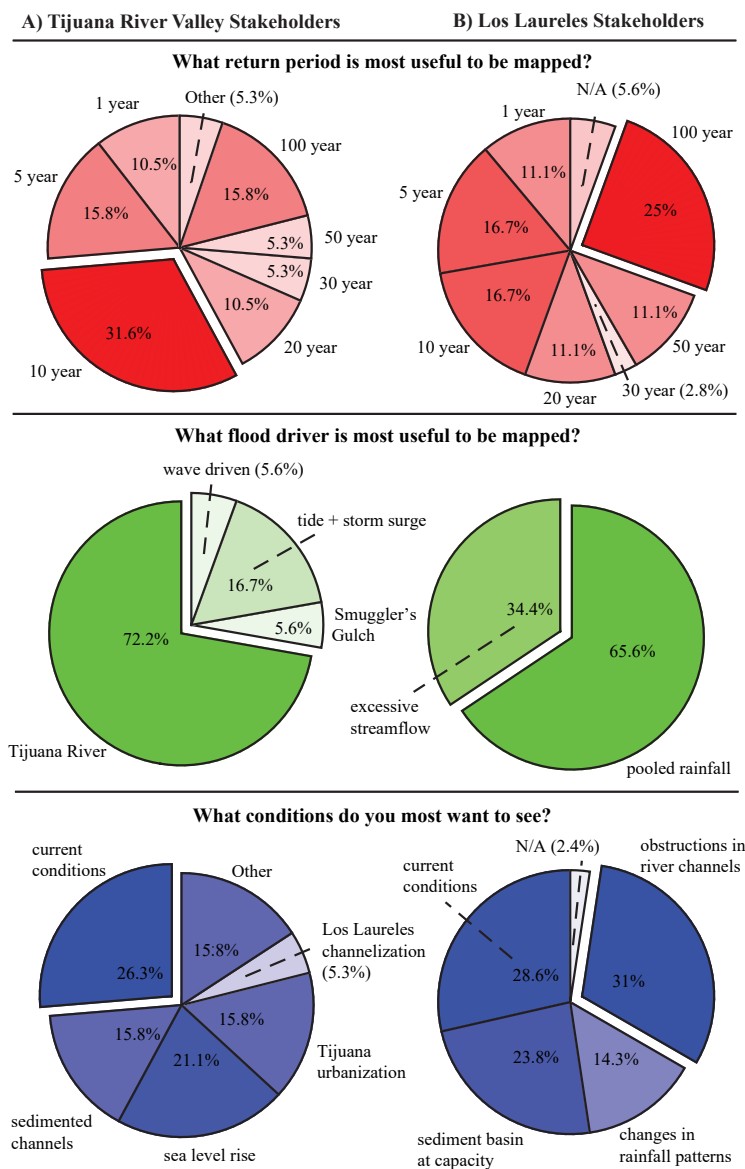

**Figure 6.** Results of the exit survey designed to provide insight on relevant flood frequencies, drivers, and environmental conditions. Column (A) includes TRV responses (22 participants), and column (B) includes LL responses (33 participants). The most distinct responses include the reported utility of maps illustrating more frequent events (TRV) and pooled rainfall (LL). The lack of a clear preference for relevant environmental conditions is worth noting. Here, "environmental conditions" are defined as the physical conditions of study area or the state of climate related variables during the simulated flood. Maps depicting possible impacts of climate change (e.g. sea level rise and changes in rainfall patterns) were not the most desired conditions for end-users of either site.



capacities and spread across the floodplain (e.g. Appendices A2-A2.4). Hence, it appears that participating end-users astutely discerned a limitation of this traditional flood modeling approach.

When asked "What conditions do you most want to see?", responses were relatively uniform among the choices offered. The exit survey revealed a slight preference for current conditions in the TRV and obstructions in river channels for LL end-
users. The transcripts further support interest in scenarios involving blockages. At both sites, end-users were concerned about trash and debris obstructing flow beneath bridges and through the culvert in LL (Fig. 1). Accordingly, end-users requested to see the impact of obstructions in river channels (Table 2), and also "worst case" flooding scenarios involving some kind of infrastructure failure. We also note that maps showing the possible impacts of climate change on flood characteristics were not especially relevant to the participating end-users. Indeed, LL participants selected "changes in rainfall patterns" least frequently
when asked "what conditions do you most want to see?", while sea level rise scenarios were slightly less preferred relative to current conditions in the TRV. Again, these results should not be viewed as representative of all end-users of flood hazard data, but it is important to note that these end-users did not view climate change impacts on flood hazards as more relevant than scenarios involving infrastructure failures or channel blockages.

By far, the most requested flood hazard data, or hydraulic model output, was for data that described the erosive potential
of flowing water. Both TRV and LL participants requested model output that can describe erosion potential or susceptibility (Table 2), and after presenting the flood force map (Fig. 4), end-users asked for the velocity of the flood waters to be related to erosion (Table 1). It is also worth noting that before being shown the flood force map, end-users at both sites requested information describing the velocity of the floodwaters. Aside from maps depicting velocities or erosion potential, end-users also requested inundation maps based on real time information or rain forecasts. One of the participants suggested a collection
of maps related to different rainfall depths as a substitute for real time mapping. Lastly, community members of the TRV requested hazard maps depicting areas susceptible to standing water. Areas susceptible to standing water are a concern among community members due to increased likelihood of mosquito activity and pollutant exposure. The following section describes both the new hazard maps and improved presentation of the hazard data that was supported by the end-user focus groups.

## 5.3   Co-produced Flood Hazard Maps

In this section, we present the flood hazard maps that were produced after re-configuring the hydraulic models based on the results of the focus groups. The maps presented herein, Fig. 7 - 9, do not include all of the maps we produced post-focus group, but rather the hazard maps that both support and expand upon previous studies offering guidance for improving flood hazard maps. All hazard maps produced in this study in response to end-user feedback can be viewed online by following the links in the *data availability* section. Herein, we also present the mapped hazard data according to the end-users' requested revisions
to exemplify effective means of communicating the mapped hazard according to these end-users.

First, we focus on end-user interest in "pooled rainfall" or storm water runoff. Figure 7 shows the pre and post-focus group flood depth maps for LL. The pre-focus group flood depth map (Fig. 7A) was produced by injecting flow hydrographs from a hydrologic model into the channel network of the hydraulic model, so the mapped flooded area corresponds to so-called *fluvial flooding*, or streamflows that exceed the capacity of the channel network (Appendix A2.4). In the post-focus group



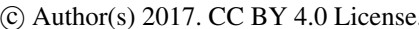



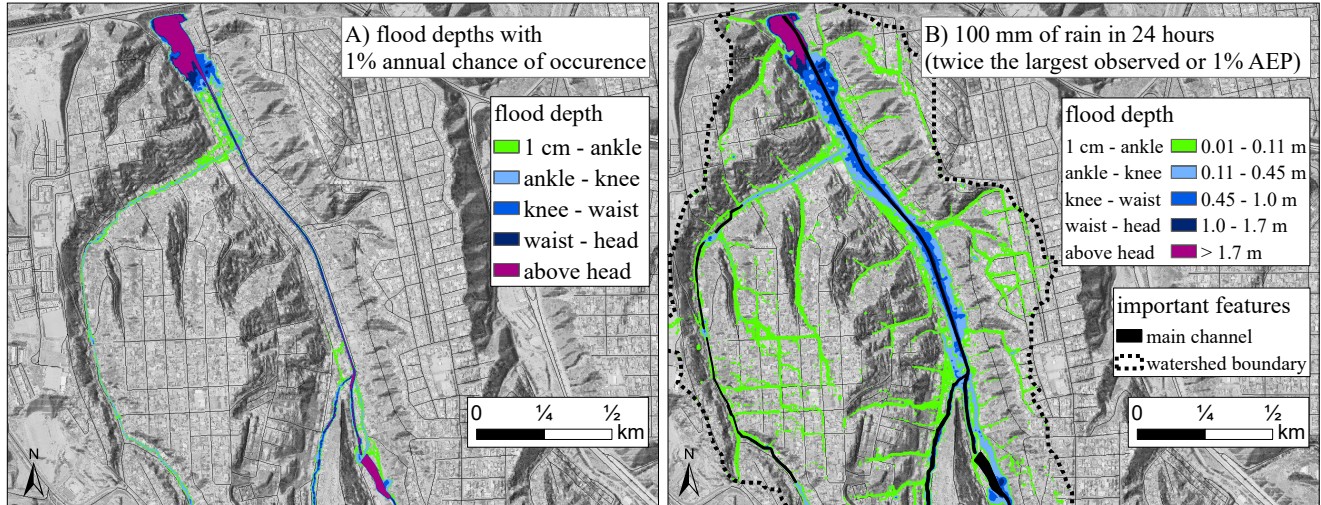

**Figure 7.** A) Pre-focus group Los Laureles flood depth map based on traditional coupling of hydrologic and hydraulic models. B) Post-focus group Los Laureles flood depth map based on routing storm-water runoff using the hydraulic model grid. Comparison between A) and B) shows that traditional hydrologic-hydraulic model coupling can misrepresent flood hazards in catchments where storm-water runoff is severe. We also present B) following the stakeholder requests. Notice that A) is described by the AEP of the flood depths, whereas B) is described by the amount of rainfall that caused the mapped hazard. The post-focus group map also includes a qualatative/quantative legend, the location of the main channel, and the watershed boundary.

flood map, on the other hand, runoff (precipitation minus infiltration) was computed for every numerical grid of the hydraulic model and routed as overland flow. Hence, the mapped flooded area corresponds to so-called *pluvial flooding*, or the combined effects of storm-water runoff and excess streamflow. Comparison between Fig. 7A and 7B reveals that the flood hazard can be significantly underestimated if only fluvial flooding is considered, showing the importance of pluvial flooding (direct storm-

water runoff) in this system.

The baseline hazard maps based on fluvial flooding did not align with stakeholders' experience, whereas the post-focus group flood hazard maps that account for pluvial flooding clearly show accumulation of floodwater along roadways and locations adjacent to the main channel. Notice that the pluvial hazard map illustrates ankle to waist deep flooding adjacent to the main channel, whereas no flooding is predicted adjacent to the main channel in the baseline map (Fig. 7). Hence, this site calls

for hydrologic modeling approaches that resolve overland flow caused by rainfall. Pluvial flood hazard studies are still fairly uncommon (Apel et al., 2016), yet there are several examples in the literature and methodological advancements are growing (Nuswantoro et al., 2016; Blanc et al., 2012; Guerreiro et al., 2017; Simões et al., 2015). Maps produced by such methods have significant potential to support FRM. For example, from an emergency management perspective, pluvial flood maps can highlight streets and low-lying areas in the floodplain that are likely to become hazardous during a severe rain storm, and from

a planning and design perspective, pluvial maps can identify areas in the floodplain with relatively poor drainage.



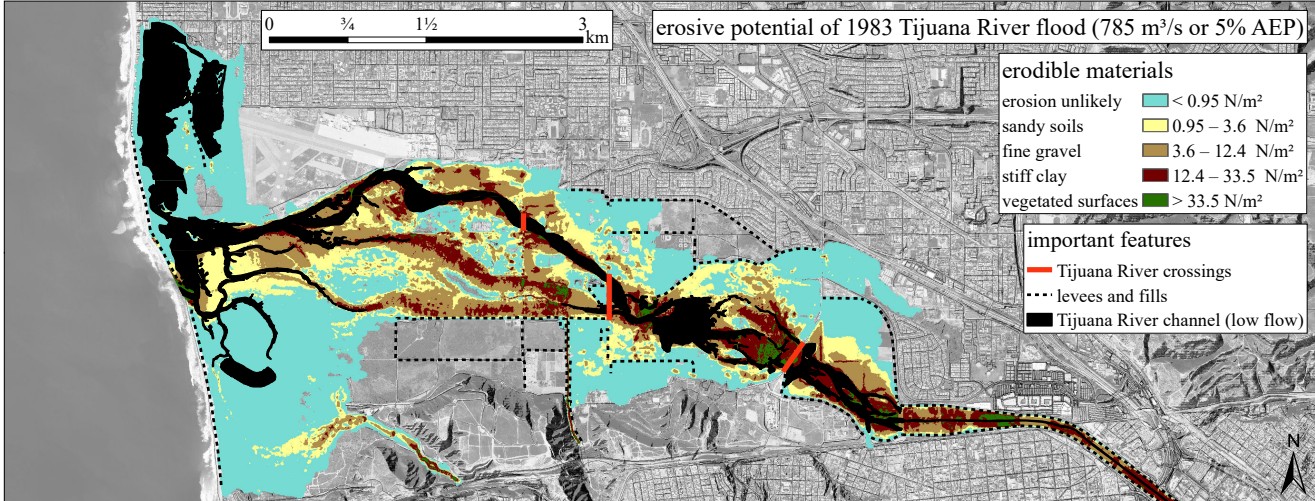

**Figure 8.** Erosion potential map of the 1983 Tijuana River flood produced for the TRV stakeholders. The contoured flood hazard variable is the maximum depth-averaged shear stress predicted by the hydraulic model during the simulated flood event. Legend thresholds were estimated from Fischenich (2001). The map also includes the locations of levees and fills, Tijuana River crossings, and the Tijuana river channel.

Notice also that the pluvial hazard map presents the data differently from the baseline map. The pre-focus group map describes the mapped hazard by the exceedance probability of the flood depths, whereas the post-focus group map describes the mapped hazard as the flood depths resulting from the corresponding rainfall depth and duration. In the latter case, the exceedance probability is included but not emphasized. This may seem like a minor difference, but end-users suggested that a description based on rainfall depth is much more useful. Rainfall depths are commonly forecast and reported publicly, so the rainfall totals are generally more relevant to end-users than frequency of occurrence or probability of exceedance. End-users' requests for the amount of rainfall and streamflow associated with the hazard map point to the need to link mapped flooding scenarios to familiar reference points.

Linking flooding scenarios to familiar reference points is difficult, however. Most flooding scenarios are defined by probabilities which are intangible to many end-users, while the significance of rainfall totals or volumetric flow rates can be unknown to users not familiar with typical site conditions. A reference point requested by end-users which has the potential to be accessible to a wide audience was the magnitude of the mapped flooding event relative to historic measurements. Notice that the pluvial flood map (Fig. 7B ) also describes the rainfall event as "twice the largest observed", which means the 1% AEP 24hr rainfall depth is roughly twice the largest recorded at the nearest rainfall gage. Describing the flooding scenario as "twice the largest observed" adds important context concerning the magnitude of the mapped event by giving the impression of a rare scenario yet within the realm of physical plausibility.

In addition to the flood hazards caused by storm-water runoff, LL and TRV stakeholders were also concerned about the erosive potential of flowing water (Table 1 and 2). Erosion potential was particularly relevant for the TRV and LL, since highly



erosive soils lead to continual dredging of flood control channels and maintenance of engineered sediment basins. Accordingly, we produced maps that illustrate the erosive potential of various floods. Figure 8 shows an example of an erosion potential map we produced for the TRV stakeholders. The erosion potential map contours the maximum depth-averaged shear stress predicted by the hydraulic model during the course of the simulated flood. We contour shear stress to illustrate erosion potential because

shear stress is commonly used to model soil erosion and design stream stabilization or restoration projects (Knapen et al., 2007; Fischenich, 2001). The qualitative scale in Fig. 8 that describes materials susceptible to erosion during the simulated flood is based on the permissible shear stresses of different stream restoration materials reported by Fischenich (2001). Notice that the units of the erosion thresholds are also included in the hazard map legend. Although it is unlikely that even technical end-users can interpret the magnitude of different shear stress values off hand (i.e. Fig. 8), the units provide the basis for the qualitative

scale and improve credibility. The shear stress values by themselves are fairly abstract, so the quantitative and qualitative scales compliment each other quite nicely.

     Natural resource managers, less often targeted as end users of flood hazard data, were an audience that sought maps of erosion potential in this study. Most flood hazard maps are tailored for public communication, strategic planning, emergency response, or insurance purposes (Meyer et al., 2012). Yet for many locations, managing land use and controlling erosion is

essential to FRM, while erosion caused by flood waters can be the most damaging aspect of a major flood (Luke et al., 2015). Maps contouring shear stress and illustrating erosion potential can be very useful for developing local land use strategies and suppressing erosion. For example, natural resource managers could restrict sensitive land uses in areas of the floodplain that are likely to experience relatively erosive flows. By overlaying the predicted shear stresses with land use/land cover data, the erosion potential map could also be used to highlight areas in need of armoring or erosion resistant vegetation.

Notice also that the example erosion potential map displays the shear stresses associated with a hindcast of the 1983 Tijuana River flood, i.e., a historic flood. Stakeholders of both sites were interested in hazard maps of historic flooding (Table 2), which we also find particularly valuable due to their relevance to end-users and ease of communication. Maps of historic floods are relevant precisely because they actually occurred; the map does not need to communicate an abstract exceedance probability or frequency. In the case of historic flood maps, the year of the flood or possibly the name of the storm can be used to describe

the flooding scenario. The 1983 Tijuana River flood was associated with a particularly strong El Nino year, which several stakeholders recall. The 1983 flood also represents a relatively frequent flood (about 20 year return period) which was deemed useful to be shown on a map by end-users of both sites (Fig. 6, Table 2).

     Lastly, we present one additional flood hazard map produced in response to community members' concerns about stagnant water in the TRV. Standing water creates breeding sites for mosquitos and a corresponding proliferation of mosquito-borne

diseases such as West Nile Virus, so it is an important public health consideration after a major flood (Kouadio et al., 2012). Figure 9 shows the depth of flood waters remaining in the TRV immediately following recession of a flood with a peak flow rate of 2300 $m^3$/s, assuming no evaporation or infiltration. Therefore, the map should be interpreted as the standing water depth immediately following the flood.

     The standing water map is unique relative to other hazard maps because it illustrates conditions immediately after rather

than during the event, and therefore represents a mechanism of supporting of *recovery* efforts - a key component of the disaster





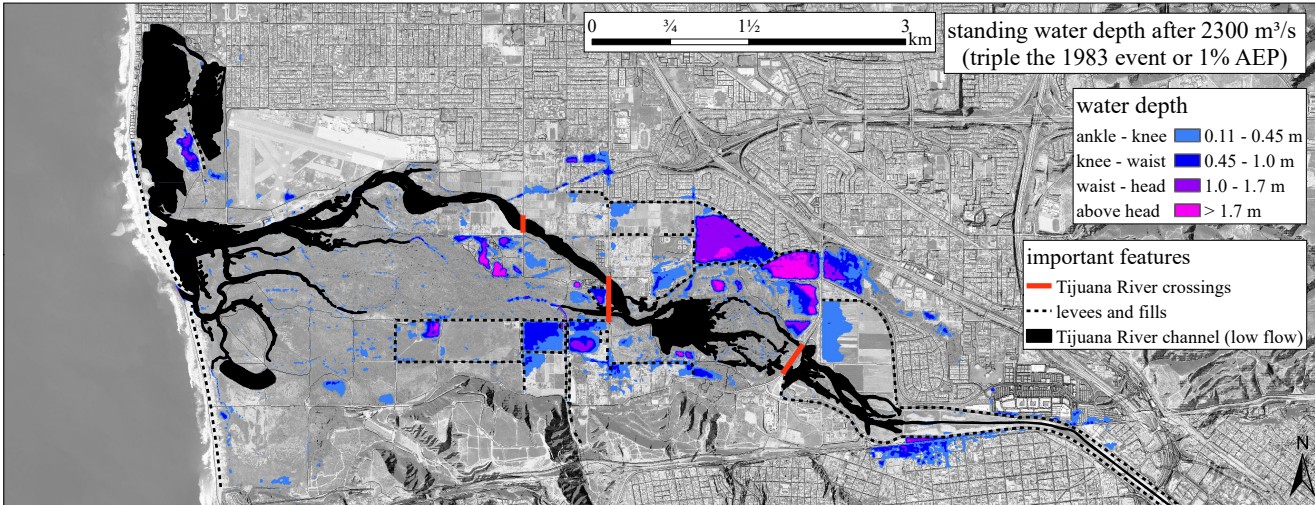

**Figure 9.** Flood hazard map illustrating pooled water that did not freely drain following a peak Tijuana River flow of 2300 m³/s, which has an estimated 1% AEP or 100 year frequency. The flood depths shown on the map are the hydraulic model solution after the hydrograph receded to baseflow conditions. The areas that do not freely drain are likely to experience increased mosquito activity and exposure to pollutants following a major flooding event.

management cycle (Khan et al., 2008). The standing water map could help authorities with mosquito abatement, deployment of pumps, and strategic positioning of emergency shelters. Additionally, a standing water map can support floodplain planning by enabling enhanced assessments of public health risks for proposed floodplain development projects.

## 6    Discussion and Recommendations

We acknowledge that the requested map scenarios and revisions of the TRV and LL end-users do not represent all end-users of flood hazard data. However, for the sake of improving the practice of flood hazard mapping, let us compare the end-users' requests herein to previous studies and flood mapping guidelines in Europe, Australia, and the US. The majority of end-user preferences from this study align with recommendations for improving flood hazard maps from European studies by Meyer et al. (2012) and Hagemeier-Klose and Wagner (2009). Concerning the presentation or communication of the mapped content,

Hagemeier-Klose and Wagner (2009) recommended that flood hazard maps should be linked to real time information such as river stage, while Bell and Tobin (2007) questioned the effectiveness of communicating floods based solely on probabilities. Both studies point to the need to link the mapped flooding scenario to concrete reference points other than frequency or exceedance probability. The need for tangible reference points was supported in this study by end-users' requests for the rainfall totals, streamflow, and relative magnitude of the mapped scenario. In flood mapping practice, however, communication

of the mapped hazard is still very much tied to probabilities.




The European handbook (Martini and Loat, 2007) states the title of the map should include the hazard parameter and probability, while the FEMA FIRM describes hazard zones by AEPs only (i.e. Fig. 2). Yet there is increasing evidence that these descriptions are ineffective. Relatively little guidance is provided regarding the map legends as well, which is another important aspect of communicating the mapped hazard. Legends are either completely described by numerical values (i.e.

depth of flooding in meters or feet) or a qualitative flood severity zone described by terms such as "low" to "severe" (FEMA, 2014). We recommend that future mapping guidance documents provide advice for different ways to communicate the mapped hazard scenario and more complete legend descriptors. Alternatives supported by this study include 1) providing qualitative and quantitative scales, and 2) describing flooding scenarios by the flood magnitude (in corresponding scientific units), the magnitude relative to an historic event, and finally the probability of the flood. The magnitude and probability of the flood

provides relevant information for technical end-users, while the magnitude related to an historic event is a tangible reference point for lay-persons. It is our opinion that *least emphasis* should be given to the probability when describing mapped flooding scenarios. Not only are concrete references preferred for describing flood risk (Bell and Tobin, 2007), but *flood probabilities and corresponding frequency* are inherently uncertain (e.g. Appendices A1.1 - A1.3, Di Baldassarre et al., 2010; Kjeldsen et al., 2014; Merwade et al., 2008; Merz and Blöschl, 2008; Stedinger and Griffis, 2008).

Regarding desirable content of flood hazard maps, stakeholder preferences from this study also align with previous work. Meyer et al. (2012) concluded that maps presenting flood hazards at different probabilities is required, velocity information should be provided when available, and the location of flood defenses and access routes should be integrated within the hazard map. All of these recommended contents were requested by the TRV and LL end-users. Hagemeier-Klose and Wagner (2009) also noted the importance of mapping more frequent events than the 1% AEP flood, which was strongly supported by the

results of the survey (Fig. 6). Thus, this study demonstrates consistency between the desired map content of end-users studied in the US, Mexico, and Europe. Flood mapping guidelines in Europe (Van Alphen et al., 2009; Martini and Loat, 2007) and Australia (AEMI, 2013) generally recommend producing this desired content. Guidelines either recommend or require the production of hazard maps associated with different probabilities and even infrastructure failure scenarios. Mapped data includes flooding extent, depths, velocities, and the depth-velocity product, while some studies even provide shear stresses

(Martini and Loat, 2007). Specific guidance is also provided for producing maps that support the FRM activities of distinct European and Australian end-user groups.

Meanwhile in the US, flood mapping guidelines are fairly extensive and standardized for producing the FEMA FIRM only (FEMA, 2016). There is a lack of guidance and direction available for producing flood hazard maps that support non-insurance aspects of FRM, such as those specifically requested by end-users in this study. The required "non-regulatory" data products of

recent FEMA Risk MAP studies (FEMA, 2014) have significant potential to support an expanded portfolio of actionable flood hazard maps in the US. For example, the required velocity grids in Risk MAP studies can be post-processed to produce erosion potential maps, while the required "flood severity grid" contains the depth-velocity data necessary for products designed to support emergency response. We recommend that future FEMA guidelines provide specific directives for producing non-regulatory flood hazard maps tailored to specific FRM objectives including land-use planning, emergency management, and

public awareness. The content of flood hazard maps should also be expanded to include pluvial flood hazards when appropriate.



Neither European, Australian, nor US flood mapping guidelines require or recommend maps characterizing pluvial flood hazard, which were of keen interest to both LL and TRV end-users. The Australian technical guidelines for engineers allude to direct rainfall models that can be used to produce pluvial hazard maps (McCowan, 2016), but because these techniques are relatively new, guidance documents do not require the production of maps depicting the pluvial hazard or intense storm-water runoff. Since techniques for estimating pluvial hazards continue to advance, formal guidelines and requirements for mapping the pluvial hazard zone should be developed. As demonstrated by end-user requests in this study - and the largely pluvial nature of the flooding disaster caused by Hurricane Harvey in the US - pluvial flooding can dominate in urban areas and needs to be considered in future mapping efforts.

## 7 Conclusions and Future Directions

Two-dimensional (2D) flood hazard models developed for the Tijuana River Valley (TRV) and Los Laureles (LL) on both sides of the US-Mexico border supported the co-development of flood hazard maps responsive to end-user management needs. 2D modeling by engineers produced a set of baseline maps that were further refined through end-user focus groups that triggered additional modeling scenarios and map revisions.

This study revealed general consistency between the mapping needs of studied end-users in the US and Mexico with those reported in European studies and guidelines published in Australia. For example, mapping requests included scenarios with different probabilities and even infrastructure failure scenarios, and end-users also requested maps of hazard variables beyond traditional flood extent, such as velocities and standing water. This study also revealed several important flood hazard mapping requests relevant to other sites:

- Flood intensity scales (e.g., depth, force or shear stress) that frame the mapped information both quantitatively and qualitatively. The quantitative scale meets end-user needs for a technical reference point, while the qualitative scale meets end-user needs to easily interpret the mapped information.

- Flood scenario descriptions that report both the magnitude of the flood in terms of rainfall or streamflow amounts and also the flood magnitude relative to an historic event. Use of concrete scenario descriptions increases the utility and relevance of mapped information across different end-users of flood hazard maps.

- Flood hazard maps that depict the erosion potential of flood waters. Erosion potential maps support end-user needs for managing sediment.

- Flood hazard maps that depict standing water following the flood. Standing water maps support recovery planning and public health concerns.

- Flood hazard maps that depict storm-water runoff or pluvial flood hazards. Baseline flood hazard maps depicted *fluvial* flooding hazards only, and after end-user focus groups revealed a deficiency in usefulness, the need for a pluvial flood





hazard modeling approach was recognized and implemented. Characterizing pluvial flood hazards is extremely important for urbanized sites with poor drainage.

Of course, the stakeholder preferences herein must be viewed cautiously, since focus group participants do not represent all end-users of flood hazard data. The primary limitation of this study is the limited number of focus group participants (55

total) and narrow geographic scope. Co-production efforts via focus groups acknowledge that community-level knowledge (and mapping preference) varies from locality-to-locality, underscoring how flood hazard knowledge should not be a "one-directional" process but an iterative learning approach that breaks down information gaps between experts and lay users in specific places - thus improving risk communication at the local level. They also produce actionable mapping information useful for reducing flood risks (Spiekermann et al., 2015; Moel et al., 2009). Indeed, restricted sample size and geographic

scope is a common caveat of flood communication and mapping preference studies (e.g Hagemeier-Klose and Wagner (2009); Meyer et al. (2012)).

In future studies, sample size limitations may be overcome by taking advantage of online information systems to present flood hazard data. Online formats also offer the opportunity for causal experiments - does the presentation of mapped hazard data make a user more (or less) likely to seek vulnerability reduction measures? This question could be answered with so called

"A/B" testing, where subjects are presented different web pages and their interactions on the web site are recorded. Our current knowledge of flood mapping preferences and hazard perceptions is based upon empirical studies with relatively small samples (Kellens et al., 2013). What can "big data" tell us about how end-users respond to, and interact with, flood hazard maps?

While online information systems offer avenues for new research, they also provide a medium for presenting an expanded portfolio of hazard maps. Relative to the EU and Australia, flood mapping practice in the US has the greatest opportunity for

expansion. Funding for flood mapping in the US remains limited (Traver, 2014), however it is relatively inexpensive to produce additional mapping products from models that are already used to produce Flood Insurance Rate Maps. Furthermore, the availability of free 2D hydraulic modeling software (HEC-RAS 5.0) and increasing abundance of metric resolution topographic data provides practitioners with the means to produce flood hazard data that was previously cost-prohibitive (Sanders, 2017). While flood mapping methods and data continue to improve - additional criteria must also be addressed to provide decision-

makers and citizens with actionable information. To be actionable, map information must help decision-makers: 1) discern vulnerability of properties from flooding; and, 2) select actions that mitigate or reduce this vulnerability (Demeritt and Nobert, 2014; McNutt, 2016; Feldman et al., 2008). By fully utilizing flood modeling technologies and mechanisms for incorporating local knowledge in the mapping process, flood hazard maps can support first responders, natural resource managers, and local residents with the information necessary to manage and respond to flood hazards.

*Data availability.* The University of California's guidelines for maintaining the privacy and confidentiality of human subjects state that data obtained from human subjects should only be accessible on a "need to know" and "minimum necessary" standard. Thus, the transcripts of focus groups conducted in this study are not publicly available. If an interested researcher wishes to review transcripts, please contact the corresponding author with 1) data requests and 2) reasoning for requesting the data.





All of the hazard maps (and several not presented herein) are available to view on an interactive system found here:

FloodRISE (2017). Tijuana River Valley Flood Hazards. University of California, Irvine. https://bit.ly/floodrise_TRV

FloodRISE (2017). Los Laureles Flood Hazards. University of California, Irvine. https://bit.ly/floodrise_GC

Data that was used to create the hazard maps includes elevation, streamflow, ocean water level, and precipitation data. The elevation data is held by the County of San Diego and could be made available via requests to the corresponding author. The streamflow, water level, and precipitation data is available here:

International Boundary and Water Commission (1960 - 2006). Flow of the Colorado River and other Western Boundary Streams and Related Data. Department of State, USA. https://ibwc.gov/Water_Data/water_bulletins.html

National Oceanic and Atmospheric Administration (1924 - 2008). Observed Water Levels at 9410230, La Jolla CA. Department of Commerce, USA. https://www.tidesandcurrents.noaa.gov/waterlevels.html?id=9410230

Department of Public Works, Flood Control Section (2003). San Diego County Hydrology Manual. County of San Diego. http://www.sandiegocounty.gov/content/dam/sdc/dpw/FLOOD_CONTROL/floodcontroldocuments/hydro-hydrologymanual.pdf

## Appendix A: Flood Hazard Mapping Methodology

The flood hazard maps presented in this study resulted from three distinct tasks: flood frequency analysis (FFA), hydrologic and hydraulic modeling, and post-processing of model output. Generally speaking, FFA estimates the recurrence interval of rare flooding events, while hydrologic and hydraulic modeling predicts the hazards associated with simulated floods (depths, velocities, extents, etc.). In this study, post-processing methods are used to combine the results of multiple simulations onto a single map. The following sections outline our FFA, hydraulic modeling, and post-processing methods so that the interested modeler can produce the hazard maps presented herein.

## A1 Flood Frequency Analysis

FFA is complicated in the coastal zone due to the multiple causes or "drivers" of flooding. In this study, we mapped flooding caused by extreme ocean levels, streamflow from the Tijuana (TJ) River, and precipitation over Los Laureles and Smuggler's gulch watersheds (Fig. 1). The presence of multiple flood drivers often warrants a multivariate approach for FFA (Salvadori and De Michele, 2013). Under this approach, multivariate extreme value analysis (EVA) is used to estimate the probability of scenarios where multiple extremes occur simultaneously. However, we did not conduct multivariate EVA in this study because of the low correlation between flood drivers and the lack of emergent flood hazards caused by the joint occurrence of extremes.

Table 1 presents the Pearson's correlation coefficient matrix between the flood drivers considered herein. The relatively low correlation is somewhat surprising but understandable. Extended periods of above average rainfall in the upper TJ River Watershed cause large streamflow events, whereas relatively short-lived coastal storm systems can elevate ocean water levels and lead to intense precipitation. The low correlation between flood drivers demonstrates that the simultaneous occurrence of extreme events would be especially rare. Perhaps more importantly, hydraulic model sensitivity analysis revealed that predicted flood depths, extents, and velocities are insensitive to the joint-occurrence of extremes in this system. For example, flood depths predicted by the hydraulic model are not sensitive to the downstream ocean level during large TJ River floods. The lack of



**Table A1.** Pearson's correlation coefficient matrix between the three drivers of flooding considered in this study. The numeric values in the table describe the correlation between the variables in the row and column headings. Correlation coefficients were determined using detrended tide gage data at La Jolla (NOAA station 9410230), precipitation measurements from the San Diego International Airport (NOAA network ID GHCND:USW00023188), and TJ River streamflow measurements at the US/MX border recorded by the International Boundary and Water Commission.

|  | Ocean Level (daily mean) | TJ Stream flow (daily mean) | Precipitation (24 hr sum) |
| --- | --- | --- | --- |
| **Ocean Level (daily mean)** | 1 | 0.12 | 0.21 |
| **TJ Stream flow (daily mean)** | 0.12 | 1 | 0.17 |
| **Precipitation (24hr sum)** | 0.21 | 0.17 | 1 |

"sufficient" correlation between drivers and the hydraulic model's insensitivity to the joint occurrence of extremes allows us to consider the flood drivers independently and use univariate EVA for frequency analysis.

### A1.1 Tijuana River Flood Frequency Analysis

FFA of TJ River flows was based on a Pearson Type III (PIII) distribution fitted to the historic record of log-transformed
annual maximum discharges. This approach is consistent with the recommended FFA methodology in the US (Hydrology Subcommittee, 1982). The data record originated from TJ River flow measurements at the US/MX border reported by the International Boundary and Water Commission. To infer the parameters of the PIII distribution, we used the Bayesian parameter estimation technique described by Luke et al. (2017) where an informative prior was used to incorporate regional information about the skewness of the PIII distribution. For the TJ River, parameter estimation was complicated by signs of nonstationarity
in the historic record, or time variant statistical properties of the annual maximum discharge data.

Figure A1A shows the full data record at the US/MX border. At the time of this study, data was not available after 2006. The *black* line in Fig. A1A denotes the year when the TJ River channelization was completed, which appeared to alter the mean and standard deviation of the flood peaks. Indeed, the pre-channelization distribution is different from the post-channelization distribution at the 0.05 significance level according to the two-sample Kolmogorov-Smirnov test (Massey Jr, 1951). Due
to the apparent change in the distribution of flood peaks following channelization, we did not use data prior to 1979 for estimation of the PIII parameters. The choice to omit data prior to channelization creates a relatively small sample size for parameter estimation and leads to large variance in the estimated return periods (Fig. A1B). Assuming stationarity following the channelization, the return periods in Fig. A1B are simply the inverse of the annual exceedance probabilities associated with the return levels on the $y$ axis. If the pre-channelization flood peaks are included in the frequency analysis, we risk bias in the
parameter estimates and resulting return periods.

Notice also that the empirical frequency curve shown in Fig. A1B appears to change shape near the 5 year return period level. We attribute this to the considerable influence of upstream reservoirs on large TJ River flows. Spillway discharges occurred during four of the annual maximum events from 1979 - 2006, but did not affect the majority of the relatively small, runoff driven annual maximum events. The various flood generating mechanisms and the change in the shape of the empirical



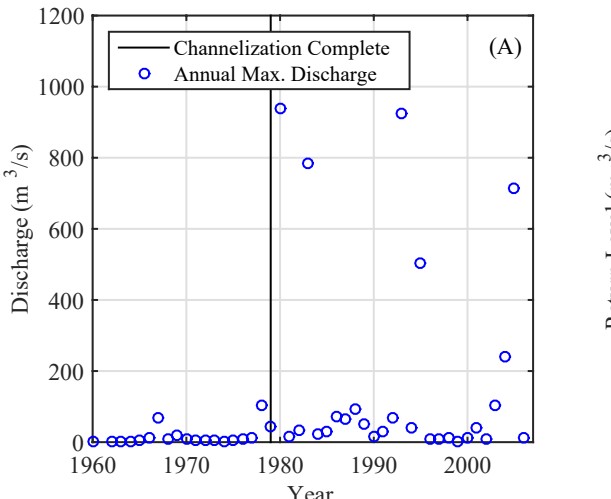 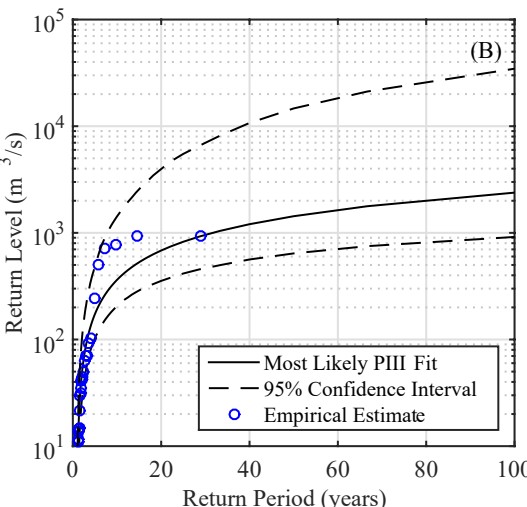

**Figure A1.** A) Annual maximum discharge record of the TJ River. Only flood peaks post-channelization were used for PIII parameter inference. B) Flood frequency estimates derived from the PIII distribution fitted to the log-transformed discharge data from 1979 - 2006. Notice the large variance in flood frequency estimates.

**Table A2.** Most likely estimates of exceedance probabilities associated with extreme ocean levels, TJ Stream flow, and precipitation. These values were used as boundary conditions for the hydrologic and hydraulic modeling.

| Annual Exceedance Probability (2015) | Ocean Level (m, NAVD88) | TJ Stream flow (m$^3$/s) | Precipitation (24 hr sum, mm) |
|---|---|---|---|
| 0.01 | 2.42 | 2333 | 101.6 |
| 0.02 | 2.40 | 1420 | 88.9 |
| 0.05 | 2.38 | 688 | 81.3 |
| 0.10 | 2.36 | 369 | 63.5 |
| 0.20 | 2.34 | 178 | 50.8 |

frequency curve both indicate that the distribution of flood peaks is not the same for small and large annual maximum events. This causes a poor fit of the PIII distribution to the data in the modern period of record and creates even more variance in the frequency estimates. It is unlikely that the variance can be significantly reduced without expanding the sample size through watershed modeling and simulation of peak flows, which was outside the scope of this study. It is very important to note that exceedance probabilities and corresponding frequency estimates based on the historic TJ River discharges alone are unavoidably uncertain.

### A1.2 Extreme Ocean Level Frequency Analysis

Extreme ocean levels near the TRV also showed signs of nonstationarity in the historic data record. Figure A2A shows the annual maximum compared to the annual mean ocean levels recorded at the La Jolla tide gage in CA, US. There is a statistically



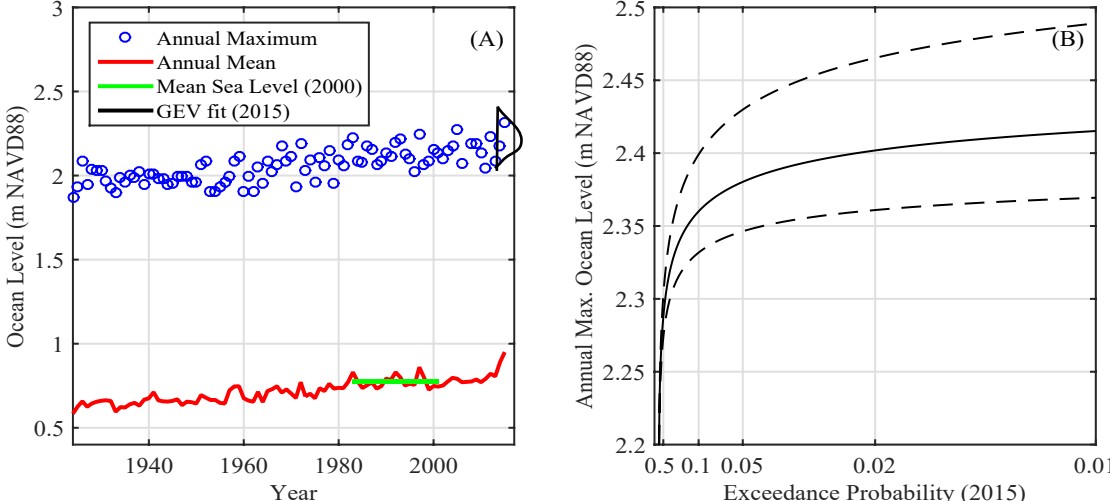

**Figure A2.** A) Annual maximum ocean levels compared to annual mean ocean levels recorded at the La Jolla tide gage in CA, USA. The *black* line shows the fitted nonstationary GEV distribution during the year 2015. B) Exceedance probabilities of extreme ocean levels derived from the 2015 GEV distribution. We do not show return periods on the $x$ axis because exceedance probabilities are expected to change as mean sea level increases.

significant trend in both the annual maximum and mean data at the 0.05 significance level, according to the Man-Kendall trend test for monotonic trends (Mann, 1945; Kendall, 1976). The persistent trend in ocean levels is not surprising, however it does complicate EVA. In this study, we explicitly modeled the change in extreme ocean levels using a nonstationary, generalized extreme value (GEV) distribution

$$X \sim \text{GEV}(\mu_t, \sigma, \xi) \tag{A1}$$

5    where the random variable $X$ is the annual maximum ocean level, and $\sigma$ and $\xi$ denote the scale and shape parameter of the GEV distribution, respectively. The time-variant location parameter, $\mu_t$, is formulated as a function of changes in mean sea level

$$\mu_t = \Delta\text{MSL}_o + \mu_o \tag{A2}$$

where $\Delta\text{MSL}_o$ is the change in annual mean sea level relative to the mean during the 1983 - 2001 tidal epoch, and $\mu_o$ is a constant off-set between the location of the GEV distribution and mean sea level. This model was proposed by Obeysekera

10    and Park (2012) to provide a method for synthesizing extreme value statistics with sea level rise scenarios. We estimated the parameters of the GEV model using Bayesian parameter inference, again with an informative prior on the shape parameter, $\xi$. The prior on $\xi$ was specified as a normal distribution centered at the La Jolla gage estimate of $\xi$ reported by Zervas (2013). Following parameter estimation, exceedance probabilities of extreme ocean levels are estimated as a function of change in mean sea level.



Figure A2B shows extreme water levels versus exceedance probabilities obtained from the fitted GEV model in the year 2015. Notice that along the $x$ axis, we no longer use return periods to describe the frequency of extreme ocean levels. The common definition of a return period relies on the assumption that exceedance probabilities are time-invariant, which is very unlikely due to anticipated changes in future mean sea level. For our hazard mapping purposes, we used the nonstationary
model to estimate the exceedance probabilities of extreme ocean levels associated with present day mean (2015) sea level. The fitted model could also be used to estimate exceedance probabilities associated with future sea levels by using sea level rise projections to define $\Delta\mathrm{MSL}_o$.

### A1.3  Precipitation Frequency Analysis

Precipitation frequency estimates over the Los Laureles and Smuggler's Gulch catchments were obtained from isopluvial maps
reported by Sholders (2003). The isopluvial maps provide 6 hour and 24 hour rainfall depths associated with different return periods. Rainfall depth and frequency estimates were taken from the isopluvial lines nearest to Smuggler's Gulch and Los Laureles catchments. To summarize and conclude the results of our frequency analysis, Table A2 includes the magnitude (return level) and exceedance probabilities for the three drivers of flooding considered herein. The values in Table A2 were used as model forcing for the hydrologic and hydraulic modeling.

## A2  Hydrologic and Hydraulic Modeling

In this study, hydrologic modeling was conducted to transform the precipitation totals over the Los Laureles and Smuggler's Gulch catchments (hereafter "the catchments") into flood hydrographs for input to the hydraulic models. Two hydraulic models were developed in this study: one covering the spatial extent of the Los Laureles Catchment, and the other including the Tijuana River Valley.

### A2.1  Hydrologic Modeling

The hydrologic models for the catchments were developed using (1) the Soil Conservation Service (SCS) curve number method to characterize precipitation losses from interception and infiltration (Ponce and Hawkins, 1996), (2) the SCS unit hydrograph method to transform excess precipitation into a hydrograph (NRCS, 1985), and (3) a 24-hour nested storm hyetograph (based on the totals in Table A2) to define the rainfall distribution within the 24-hour simulation (Sholders, 2003). The channel
flow within the catchments was routed between sub-basins using the kinematic wave model described by (USACE, 2000). Watershed areas, channel geometries, and basin slopes were estimated from a digital elevation model (DEM) with a 0.76 m (2.5 ft) horizontal resolution, which originated from a 2014 liDar Survey conducted by the County of San Diego. Curve numbers were defined based on land use data from the University of Arizona Remote Sensing Center and literature values from USACE, 2000. Unfortunately, flow measurements within or at the catchment outlets were not available at the time of the
study, so hydrologic model calibration was not possible.





## A2.2   Los Laureles Hydraulic Model

Flows in Los Laureles were routed using BreZo (Sanders et al., 2010; Kim et al., 2014), which solves the shallow water equations using a 2D finite volume scheme optimized for applications involving natural topography. BreZo operates on an unstructured grid of triangular or quadrilateral cells, which allows for variable mesh resolution and geometries throughout the

modeling domain. The Los Laureles modeling domain covers the entire area of the Los Laureles Watershed (Fig. 1), with an average cell area of 13.4 m$^2$. The Los Laureles mesh was generated using Gmsh (Geuzaine and Remacle, 2009) to create a structured, quadrilateral grid along channels and a mixed-mesh of triangular and quadrilateral cells in the floodplain. The structured, quadrilateral portion of the mesh was aligned with trapezoidal channels and small gutters along streets within Los Laureles. We used GPS measurements of channel bank and bottom elevations to define the elevation of mesh nodes aligned

with channels. Mesh node elevations within the floodplain were based on the DEM from the 2014 liDAR Survey. Resistance was characterized using spatially-varying Manning's n values, where a value of 0.015 s/m$^{(1/3)}$ was used for concrete surfaces, and 0.035 s/m$^{(1/3)}$ was used for natural areas of the floodplain. Again, no flow or stage measurements existed within Los Laureles at the time of the study, so the hydraulic model is un-calibrated.

## A2.3   Tijuana River Valley Hydraulic Model

Tijuana River Valley (TRV) flows were also routed using BreZo (Sanders et al., 2010; Kim et al., 2014). The TRV mesh was generated using Triangle (Shewchuk, 1996), resulting in a triangular mesh of variable resolution throughout the modeling domain. The mesh domain is bounded by the Pacific ocean to the West, Imperial Beach to the North, and the elevated terrain near the US/MX border to the South (Fig. 1). Mesh edges were aligned with the TJ River channel banks and small levee systems found within the TRV. The resolution of the mesh is highly variable; cells overlapping small channels in the Estuary

were assigned an area of 36 m$^2$, whereas relatively homogeneous regions in the floodplain were assigned a cell area of 100 m$^2$. Mesh node elevations within the floodplain were also based on the DEM from the 2014 liDAR Survey. Flow resistance was characterized using spatially varying Manning's n values, where the Manning's n value was determined based on land use data. Values ranged from 0.011 s/m$^{(1/3)}$ for the TJ River channel, to 0.1 s/m$^{(1/3)}$ for densely vegetated, riparian areas.

The TRV hydraulic model was validated using observations of water surface elevations in the Estuary and the TJ River at the

US/MX border. Observations of Estuary water levels and TJ River stage were obtained from the National Estuarine Research Reserve System and the International Boundary and Water Commission, respectively. In the TJ River channel, comparison of modeled stage to observed stage yielded a root mean square error of 0.25 m for TJ river flow rates of 0 - 1040 m$^3$/s. The error in modeled water surface elevations is most likely due differences in the sediment level in the TJ channel between the observed and modeled events. To validate the downstream region of the TRV model, we simulated a 2-week tidal cycle at the ocean

boundary and compared modeled water surface elevations to those observed during the same 2-week period. Over the 2 week period, the root mean square error between observed and modeled water surface elevations was 0.07 m. The error in modeled water surface elevations in the Estuary is within the error of the topographic data. Thus, the TRV hydraulic model accurately reproduces water surface elevations for both riverine and tidal forcing.





### A2.4 Hydraulic Model Forcing

Developing hydraulic modeling scenarios appropriate for hazard mapping requires careful consideration. For the TRV hydraulic model, the presence of multiple flood drivers complicates the development of scenarios that represent the exceedance probabilities in Table A2. This is not an issue for the Los Laureles hydraulic model, since only one driver of flooding was considered in Los Laureles. From a hazard mapping perspective, the challenge in Los Laureles is coupling the hydrologic model with the hydraulic model. Our approaches for addressing both of these issues are outlined in this section.

In this study, the Los Laureles hydraulic model was coupled with the hydrologic model differently before and after end-user focus groups. Prior to the end-user focus groups, the hydrographs generated by the hydrologic model described in Section A2 were input to the Los Laureles hydraulic model as point-sources of discharge at the sub-watershed outlets within the hydrologic modeling domain. Under this approach, all effective runoff reaches the storm water channels without explicit flow routing in the out-of-bank areas. As discussed in Section 5.3, this is not ideal for hazard mapping because only areas susceptible to channel overtopping appear on the hazard map. After end-user focus groups, we added the effective precipitation directly to the 2D modeling grid. Since the 2D modeling grid covers the entire Los Laureles catchment area, there was no need to add fluvial discharges to the modeling domain using boundary conditions or point-sources. We used the SCS curve number method to estimate the effective precipitation from the rainfall hyetographs, where each 2D model cell was assigned a curve number based on land-use. The effective rainfall hyetographs were added to the 2D grid as spatially distributed source of discharge. Flow was routed for partially wet cells using kinematic wave theory with the friction slope approximated using Manning's equation, whereas flow was routed using the 2D shallow water equations for fully-wetted cells. Explicit routing of overland flow results in the hazard maps similar to those shown in Fig. 7, which we consider more useful based upon the results of our end-user focus groups.

To address the issue of multiple flood drivers in the TRV, we simulated the extreme conditions of each driver separately with the TRV hydraulic model. It is important to note that if multiple extremes are modeled simultaneously, for example a scenario where an abnormal ocean level coincides with an extreme TJ River flood, the resulting flood hazard would not be associated with the exceedance probabilities of the individual events. Thus, during the TJ River flood simulations, the downstream (ocean) boundary conditions were defined as mean-tidal cycles, and the flows from the catchments were set to zero. To define the the TJ River flow hydrographs, we scaled the hydrograph associated with the 1980 flood to the peak discharges in Table A2. These scaled hydrographs served as boundary conditions at the upstream boundary of the modeling domain. The resulting TJ river flood hazards predicted by the hydraulic model are associated with the exceedance probabilities defined in Table A2. The extreme ocean level simulations were developed using the same reasoning. An average, 12-hour tidal cycle was scaled to the extreme ocean levels in Table A2 to define the ocean boundary of the model. During the extreme ocean level simulations, flows from the catchments and the TJ River were set to zero. Lastly, for the extreme precipitation scenarios, the hydrographs predicted by the hydrologic model associated with the rainfall events in Table A2 were input as point sources to the hydraulic model at the catchment outlets (Fig. 1). During these simulations, the ocean boundary conditions were defined as mean-tidal cycles, while the flows from the TJ River were set to zero. This approach results in an ensemble of hydraulic model output that





is a function of 1) exceedance probability and 2) flood drivers. We combined the results of these simulations into hazard maps using probability rules and post-processing techniques.

## A3 Post-Processing Methods

For each hydraulic model simulation, we saved the cell-centered maximum flood depths, unit discharges, depth averaged shear
stresses, and durations of depth greater than 0.11 m. These "hazard variables", denoted collectively as $H$, were processed following simulation to produce the various hazard maps. To create a continuous raster surface from the discrete $H$ values of the hydraulic model cell-centers, we used an inverse distance weighted interpolation scheme. The continuous raster surfaces are the mapped hazard data shown in this study. The Los Laureles hazard maps of $H$ required no further post-processing, since only one driver was considered. However, for the TRV hazard maps, we contoured the maximum value between the three
different drivers of flooding

$$\mathbb{H}_i = \max(\{H_i^A,\ H_i^B,\ H_i^C\}) \tag{A3}$$

where $\mathbb{H}_i$ is the mapped hazard value at raster surface location $i$, and the superscripts $A$, $B$, and $C$ denote $H$ values resulting from extreme ocean level, TJ river flow, and extreme precipitation simulations, respectively. $H_i^A$, $H_i^B$, and $H_i^C$ are associated with the same exceedance probabilities when combined in this manner. The TRV hazard maps therefore depict flood hazards with specific exceedance probabilities resulting from either driver of flooding considered, depending on location within the
TRV.

The maps contouring the exceedance probabilities of specific flood hazard thresholds required additional processing. First, raster surfaces of exceedance probability $P_i$ were created for each flood driver considered. Given a set of hydraulic model output corresponding to $n$ exceedance probabilities, $p_1, p_2, \ldots p_n$, the exceedance probability at raster location $i$ is given by the largest value of $p$ for which the hazard level $H_i$ exceeds a prescribed threshold. To account for the three drivers of flooding in
TRV, three probability rasters surfaces were computed: $P_i^A$, $P_i^B$ and $P_i^C$, which denote the probability of exceeding the hazard threshold from extreme ocean levels, TJ river floods, and extreme precipitation, respectively. Next, based on the assumption of independence between drivers, the mapped probability is given by

$$\mathbb{P}_i = P_i^A + P_i^B + P_i^C - P_i^A \cdot P_i^B - P_i^A \cdot P_i^C - P_i^B \cdot P_i^C - P_i^A \cdot P_i^B \cdot P_i^C \tag{A4}$$

where $\mathbb{P}_i$ is the exceedance probability of the hazard threshold at location $i$ resulting from all drivers of flooding considered. In Los Laureles, $\mathbb{P}_i = P_i^C$ since only flooding caused by extreme precipitation was simulated. Notice that equation A4 results
from probability addition rules of three independent events, and could be expanded or contracted depending on the number of flood drivers considered. Figure A3 illustrates this mapping methodology.

*Author contributions.* Adam Luke developed the hydraulic and hydrologic models, produced all flood hazard maps, participated in focus group design and implementation, analyzed and coded focus group transcripts, and prepared the manuscript with contributions from all co-authors. Brett Sanders, Jochen Schubert, and Amir AghaKouchak guided hydraulic and hydrologic model development and hazard mapping.




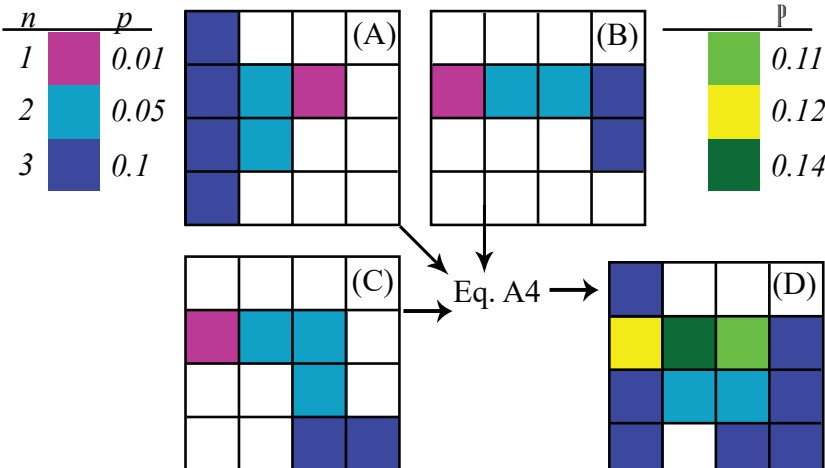

**Figure A3.** Illustration of methodology for mapping the exceedance probability of a hazard threshold from multiple flood drivers. A) Raster of $P_i^A$ values for generic flood driver, $A$. B) Raster of $P_i^B$ values for generic flood driver, $B$. C) Raster of $P_i^C$ values for generic flood driver, $C$. D) Exceedance probability raster of $\mathbb{P}_i$ values resulting from flood driver $A$, $B$, or $C$.

Kristen Goodrich analyzed and coded focus group transcripts and also guided focus group design and implementation. Dave Feldman, Danielle Boudreau, Ana Eguiarte, Kimberly Serrano, Abigail Reyes, Victoria Basolo, and Richard Matthew guided focus group design and implementation.

*Competing interests.* The authors declare they have no conflicts of interest

5  *Acknowledgements.* This work was made possible by a grant from the United States National Science Foundation (grant DMS-1331611), whose support we gratefully acknowledge. We also thankfully acknowledge Trent Biggs and Kristine Taniguchi for providing channel geometry data in the Los Laureles catchment. To all focus group participants, we thank you for your willingness to participate and the insights learned from your contributions.



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
