# Peer review of "Going beyond the Flood Insurance Rate Map: insights from flood hazard map co-production"

_Natural Hazards and Earth System Sciences, 2017_

## Referee Comment (RC1) · S. Fuchs (Referee) · 4 Nov 2017

Review on nhess-2017-384 (Going beyond the Flood Insurance Rate Map: insights from flood hazard map co-production) by Luke et al.

Taking US flood risk maps as an example, the authors assess the need to deepen our understanding about factors that make such maps useful and understandable for local end-users, an emerging issue not only in the US but world-wide. Therefore, the topic is of considerable scientific interest and high practical relevance; a paper dealing with this topic is within the scope of NHESS and shall definitely be published.

The article is generally well-written, methodological sound and the methods are accordingly mirrored by the results. I only have minor comments that should be addressed by the authors:

- On page 2, lines 2 ff. the authors state that "Insured losses from natural disasters have been increasing globally (Munich Re, 2005), largely from the growing exposure and value of vulnerable assets (Bouwer, 2011)." – The authors should be aware that this is generally undoubtable, however, exposure (and associated vulnerability) is subject to considerable spatial (and temporal) variation, as for example shown for European mountain regions by Fuchs et al. (2015; 2017) [and please be aware that I am not providing these sources to press you for more citations, which would be against good scientific practice and is not in line with the rules of NHESS]. From my point of view it is just important to be a bit careful with these general statements since the question of growing exposure is a tricky one in areas with limited development space, and given certain political incentives for land development.

- The authors may wish to access the EU flood directive in more detail. As stated on page 2, lines 20 ff., they argue that "In the European Union (EU), member countries are under a mandate to develop national flood hazard maps, and general guidelines for meeting enduser needs have been developed based on participatory processes". In contrast, the EU Floods Directive explicitly focuses on flood RISK maps (on various scales and focusing on different hazard scenarios), leading finally to flood risk management plans. Therefore, it is not only the hazard information that should be communicated, but information on risk. The Directive is attached as a supplement.

- Authors should carefully check their reference list; multiple-author sources are cited differently.

In general, the results are in line with those from European studies, and again show how challenging the topic is. I encourage the authors to further develop their studies, and as these have significant potential to support an expanded portfolio of flood risk maps.

References:

Commission of the European Communities: Directive 2007/60/EC of the European Parliament and of the Council of 23 October 2007 on the assessment and management of flood risks, Official Journal of the European Union, L 288, 27-34, 2007.

Fuchs, S., Keiler, M., and Zischg, A.: A spatiotemporal multi-hazard exposure assessment based on property data, Natural Hazards and Earth System Sciences, 15, 2127-2142, 2015.

Fuchs, S., Röthlisberger, V., Thaler, T., Zischg, A., and Keiler, M.: Natural hazard management from a coevolutionary perspective: Exposure and policy response in the European Alps, Annals of the American Association of Geographers, 107, 382-392, 2017.

Please also note the supplement to this comment:
https://www.nat-hazards-earth-syst-sci-discuss.net/nhess-2017-384/nhess-2017-384-RC1-supplement.pdf

---

## Referee Comment (RC2) · F. Dottori (Referee) · 30 Dec 2017

This work addresses important topics that are definitely of interest for the readers of NHESS. Generally speaking, the article is well written, the methodology is sound and presented with the necessary detail and the findings are well presented.

My only (moderate) remark is the lack of a discussion regarding the uncertainty of flood hazard maps and how uncertainty can be communicated to end-users. At page 20, lines 10-12, the Authors state that "... flood probabilities and corresponding frequency are inherently uncertain...". While I fully agree with such a statement, the same can be said for all the hazard variables included in flood hazard maps. To give an example: even if the use of historical flood events as reference may reduce uncertainty,

not all the original boundary conditions may be determined with precision (e.g. river channel morphology or rainfall distribution). Please note that this is not a criticism to the methodology, which is in my opinion up to the current state of the art. However, it would be interesting to know how the accuracy (and the uncertainty) of the hazard maps is perceived by end-users: For instance, what is the precision assumed by end users for the numerical variables (e.g. +/- 10 cm for flood depths)? Does this value agree with the precision expected by the Authors? How did the Authors communicated the assumptions used for flood simulations? If these topics were not addressed within the focus groups, maybe the authors could still include them in the discussion.

Minor comments

Section 2: what is the extent of the two areas analyzed in the paper? Does the extent correspond to the areas shown in Figure 1A and 1C, or are these just a sample of the areas?

Page 21, line 1: This is not completely correct. Even if pluvial flood hazard is not explicitly mentioned in the EU Floods Directive, several European countries did include pluvial floods in their national risk assessment, as it was considered a relevant component of the overall flood risk. For more details, please see the reports regarding the status of the implementation of the Floods Directive and available here: http://ec.europa.eu/environment/water/flood_risk/overview.htm

---

## Author Comment (AC1) · 5 Jan 2018

Comment 1: On page 2, lines 2 ff. the authors state that "Insured losses from natural disasters have been increasing globally (Munich Re, 2005), largely from the growing exposure and value of vulnerable assets (Bouwer, 2011)." – The authors should be aware that this is generally undoubtable, however, exposure (and associated vulnerability) is subject to considerable spatial (and temporal) variation, as for example shown for European mountain regions by Fuchs et al. (2015; 2017) [and please be aware that I am not providing these sources to press you for more citations, which would be against good scientific practice and is not in line with the rules of NHESS]. From my point of view it is just important to be a bit careful with these general statements since the question of growing exposure is a tricky one in areas with limited development

space, and given certain political incentives for land development.

Author's Response: Thank you for your perspective on the complex issue of growing losses from natural disasters. In the revised version, we acknowledge that the cause of growing exposure is not simple:

Author's changes to manuscript: Insured losses from natural disasters have increased globally (Munich Re, 2005), and while the causes of growing losses are complex and debatable, the increasing exposure and value of capital at risk has undoubtedly played a major role (Bouwer, 2011). Exposure to flooding is particularly acute in the United States (US), where a combination of subsidized flood insurance and homeowner tax incentive has actually encouraged risky development in floodplains and coastal zones (Bagstad et al. 2007).

Comment 2: - The authors may wish to access the EU flood directive in more detail. As stated on page 2, lines 20 ff., they argue that "In the European Union (EU), member countries are under a mandate to develop national flood hazard maps, and general guidelines for meeting enduser needs have been developed based on participatory processes". In contrast, the EU Floods Directive explicitly focuses on flood RISK maps (on various scales and focusing on different hazard scenarios), leading finally to flood risk management plans. Therefore, it is not only the hazard information that should be communicated, but information on risk. The Directive is attached as a supplement.

Author's response: Thank you for providing the Directive – we will address the purpose of the Directive more explicitly in the revised version:

Author's changes to manuscript: Flood hazard maps are the most commonly used tool for flood risk communication and management. In the European Union (EU), member countries are under a mandate to develop national flood hazard maps, flood risk maps, and FRM plans based upon the mapped information (Council of the European Union, 2007).

Comment 3: - Authors should carefully check their reference list; multiple-author sources are cited differently Author's Response

Bibliography has been updated to remove multiple entries of " Flood maps in Europe-methods, availability, and use".

Author's References:

Munich Re: Topics Geo Annual review: natural catastrophes 2005, Munich Re, Munich, 15, 2005.

Bouwer, L. M.: Have disaster losses increased due to anthropogenic climate change?, Bulletin of the American Meteorological Society, 92,39–46, 2011.

Bagstad, K. J., Stapleton, K., and D'Agostino, J. R.: Taxes, subsidies, and insurance as drivers of United States coastal development, Ecological Economics, 63, 285–298, 2007.

Council of European Union: Council Directive 2007/60/EC on the assessment and management of flood risks, OJ L 288, 2007.

---

## Author Comment (AC2) · 5 Jan 2018

Comment 1: My only (moderate) remark is the lack of a discussion regarding the uncertainty of flood hazard maps and how uncertainty can be communicated to end-users. At page 20, lines 10-12, the Authors state that "... flood probabilities and corresponding frequency are inherently uncertain...". While I fully agree with such a statement, the same can be said for all the hazard variables included in flood hazard maps. To give an example: even if the use of historical flood events as reference may reduce uncertainty, not all the original boundary conditions may be determined with precision (e.g. river channel morphology or rainfall distribution). Please note that this is not a criticism to the methodology, which is in my opinion up to the current state of the art. However, it would be interesting to know how the accuracy (and the uncertainty) of the hazard

maps is perceived by end-users: For instance, what is the precision assumed by end users for the numerical variables (e.g. +/- 10 cm for flood depths)? Does this value agree with the precision expected by the Authors? How did the Authors communicated the assumptions used for flood simulations? If these topics were not addressed within the focus groups, maybe the authors could still include them in the discussion.

Author's response: Thank you for commenting on this important issue. While we did not explicitly address this issue in the focus groups, the discussion now includes a paragraph on uncertainty and the conclusion section has been updated as follows:

Author's Changes to Manuscript:

Expanded discussion (insert at line 14, page 20 of original submission): Indeed, all hazard variables illustrated in flood maps are inherently uncertain, however it is remarkable that perhaps the most uncertain and complex characteristic of floods is also the primary descriptor.

Uncertainties associated with flood mapping products are rarely quantified let alone communicated, and in this study, we did not address the important issue of communicating uncertainty in flood maps to end-users. In one of the few studies that has explicitly addressed communicating uncertainty in the FEMA FIRMs' floodplain boundaries, Soden et al. (2017) showed that providing end-users with contrasting information (i.e. the 1% AEP flood extent versus an observed flooding extent) led to important flood hazard discourse and curiosity regarding flood mapping methodology. While it may seem counterproductive to purposefully expose the limitations of floodplain delineation, such innovative communication strategies force end-users to confront the deterministic standards that our institutions require for regulatory purposes. Confrontation with the limits of science promotes contemplation and is certainly worth further investigation in the context of flood hazard mapping and communication.

Reference to uncertainty in conclusions (insert at line 15, page 22 of original submission): Online formats offer the opportunity for causal experiments - do different hazard

variables make a user more (or less) likely to seek vulnerability reduction measures? How do different presentations of uncertainty in mapped data influence end-users' desire to seek further information? These questions could be answered with so called "A/B" testing, where subjects are presented different web pages and their interactions on the web site are recorded.

Comment 2: What is the extent of the two areas analyzed in the paper? Does the extent correspond to the areas shown in Figure 1A and 1C, or are these just a sample of the areas?

Author's Response: The extent of the area analyzed in the paper includes the Los Laureles catchment in Figure 1A and the Tijuana River Valley shown in Figure 1C.

Comment 3 This is not completely correct. Even if pluvial flood hazard is not explicitly mentioned in the EU Floods Directive, several European countries did include pluvial floods in their national risk assessment, as it was considered a relevant component of the overall flood risk. For more details, please see the reports regarding the status of the implementation of the Floods Directive and available here: http://ec.europa.eu/environment/water/flood_risk/overview.htm

Author's Response: Thank you for bringing this to our attention!

Author's Changes to Manuscript (insert at line 2, page 21): Neither European, Australian, nor US flood mapping guidelines explicitly require or recommend maps characterizing pluvial flood hazard, which were of keen interest to both LL and TRV end-users. However, many EU member states have included pluvial flood hazard assessments in response to the Floods Directive (Nixon et al., 2015). The Australian technical guidelines for engineers allude to direct rainfall models that can be used to produce pluvial hazard maps (McCowan, 2016), but because these techniques are relatively new, guidance documents do not require the production of maps depicting the pluvial hazard or intense storm-water runoff. Since techniques for estimating pluvial hazards continue to advance, formal guidelines and requirements for mapping the pluvial hazard zone

should be developed, especially in the US. As demonstrated by end-user requests in this study - and the largely pluvial nature of the flooding disaster caused by Hurricane Harvey in the US - pluvial flooding can dominate in urban areas and needs to be considered in future mapping efforts.

Author's references:

Soden, R., Sprain, L., and Palen, L.: Thin Grey Lines: Confrontations With Risk on Colorado's Front Range, in: Proceedings of the 2017 10 CHI Conference on Human Factors in Computing Systems, pp. 2042–2053, ACM, 2017.

Nixon, S., Horn, J., Hödl-Kreuzbauer, E., Harmsel, A. t., Erdeghem, D. V., and Dworak, T.: European Overview Assessment of Member States' reports on Preliminary Flood Risk Assessment and Indentification of Areas of Potentially Significant Flood Risk, Tech. rep., European Union, 2015.

McCowan, A.: Flood Hydraulics: Numerical Models, Chapter 4 of Book 6 in Australian Rainfall and Runoff: A Guide to Flood Estimation, Commonwealth of Australia, 2016.